# Impact of mid Eocene greenhouse warming on America's southernmost floras

Damián A. Fernández[1,2,3], Luis Palazzesi ● [3,4 ✉], M. Sol González Estebenet[3,5], M. Cristina Tellería[3,6] &
Viviana D. Barreda[3,4]

A major climate shift took place about 40 Myr ago—the Middle Eocene Climatic Optimum or MECO—triggered by a significant rise of atmospheric $CO_2$ concentrations. The biotic response to this MECO is well documented in the marine realm, but poorly explored in adjacent landmasses. Here, we quantify the response of the floras from America's southernmost latitudes based on the analysis of terrestrially derived spores and pollen grains from the mid-late Eocene (~46–34 Myr) of southern Patagonia. Robust nonparametric estimators indicate that floras in southern Patagonia were in average ~40% more diverse during the MECO than pre-MECO and post-MECO intervals. The high atmospheric $CO_2$ and increasing temperatures may have favored the combination of neotropical migrants with Gondwanan species, explaining in part the high diversity that we observed during the MECO. Our reconstructed biota reflects a greenhouse world and offers a climatic and ecological deep time scenario of an ice-free sub-Antarctic realm.

[1] Laboratorio de Geomorfología y Cuaternario, CADIC, Ushuaia, Argentina. [2] Instituto de Ciencias Polares, Ambiente y Recursos Naturales, Universidad Nacional de Tierra del Fuego, Ushuaia, Argentina. [3] Consejo Nacional de Investigaciones Científicas y Técnicas (CONICET), Buenos Aires, Argentina. [4] Sección Paleopalinología, Museo Argentino de Ciencias Naturales "Bernardino Rivadavia", Buenos Aires, Argentina. [5] Instituto Geológico del Sur (INGEOSUR), Universidad Nacional del Sur, Departamento de Geología, Bahía Blanca, Argentina. [6] Laboratorio de Sistemática y Biología Evolutiva, Museo de La Plata, La Plata, Argentina. ✉email: lpalazzesi@macn.gov.ar

The Earth has undergone a general cooling trend for the past ~50 Myr, culminating in a continental-scale glaciation of Antarctica at the Eocene–Oligocene boundary. The Middle Eocene Climatic Optimum (or MECO) occurred about 40 million years ago, interrupting that cooling trend when vast amounts of $CO_2$ were injected into the atmosphere, and sea surface temperature increased as much as 6 °C[1]. This warming event—widely recognized by a prominent perturbation in both oxygen and carbon stable isotopes—lasted about 500–600 Kyr[2,3].

Ecological models can potentially predict the impact of species diversity to rising temperatures and atmospheric $CO_2$. However, only the fossil record provides empirical evidence on how biodiversity is affected by long-term climatic transitions, even during global warming events. For example, fossil floras are known to have peaked in diversity during earlier hyperthermal episodes either at low[4] or high[5] paleo-latitudes of the American continent. The MECO may have also influenced terrestrial biotas, yet the magnitude of this response remains largely unknown as most published data have traditionally focused on the marine realm; it is still unclear whether biotic diversity increased, whether turnovers were gradual or step-like or whether tropical immigrants were frequent at the highest latitudes during the MECO.

Here, we quantitatively estimate shifts in floristic diversity on the basis of the analysis of terrestrially derived spore-pollen assemblages preserved in well-constrained marine Patagonian deposits (Río Turbio Formation) encompassing the MECO as well as the pre- and post-MECO. We used the dinocyst (i.e. dinoflagellate cyst) record to constrain the age of the spore and pollen bearing sediments. Our study reinforces the importance of the fossil spore-pollen record to explore past diversity trends and represents a new explicit picture of how floras responded to a greenhouse event in America's highest austral latitudes.

## Results

We recovered well preserved palynomorphs (i.e. dinocysts, spores, and pollen grains) in 53 samples of the Río Turbio Formation, southern South America (Fig. 1; Supplementary Fig. 1). From those, we selected 32 samples based on lithology (e.g. coal seams were removed from the analysis) and paleoenvironmental settings (see Methods; Supplementary Note 1). The dinocyst species are given in Supplementary Data 1 and Supplementary Fig. 2. We detected three major groups of samples based on our cluster analysis using dinocyst frequency (Supplementary Data 1; Supplementary Fig. 3), probably driven by shifts in the most frequent species through the composite section: *Enneadocysta dictyostila*. This is a key species of the Middle Eocene Climatic Optimum (MECO) in the southernmost latitudes (see Supplementary Note 2). The three groups of samples detected by our cluster analysis represent distinct time intervals: Interval (A)

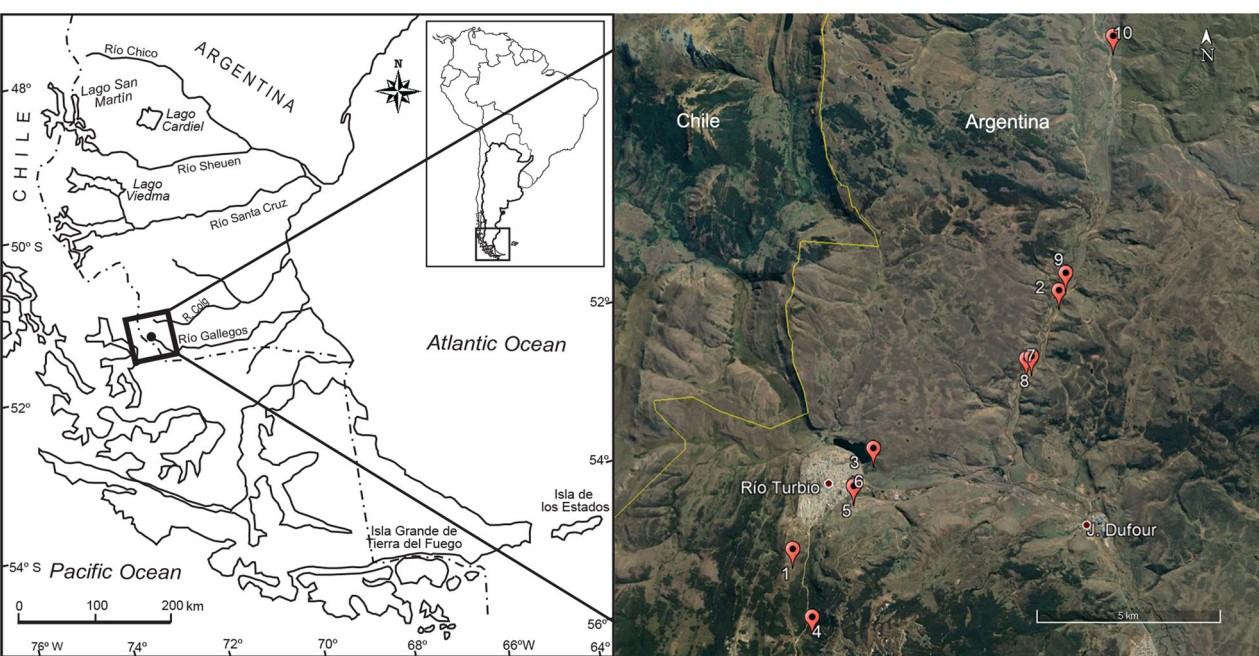

**Fig. 1 Study area.** Location map showing distribution of Eocene sedimentary rocks of the Río Turbio Formation, Santa Cruz Province, Patagonia, southern South America.

**Table 1 Diversity estimators derived from spore-pollen records from southern Patagonia.**

| Samples | Age (Mya) | Interval | Evenness | | EWSD (size = 250) | | EWSD (cov = 0.8) | | EWSD (Chao1) | |
|---|---|---|---|---|---|---|---|---|---|---|
| 26–32 | <36 | Post-MECO | 0.347 | 0.347 | 26.661 | 26.661 | 13.184 | 13.184 | 35.184 | 35.184 |
| 22–25 | 41–39 | MECO | 0.382 | 0.364 | 35.904 | 35.855 | 18.419 | 17.263 | 54.874 | 57.714 |
| 13–21 | | | 0.364 | 0.364 | 36.305 | 35.855 | 16.967 | 17.263 | 57.845 | 57.714 |
| 8–12 | | | 0.346 | 0.364 | 35.357 | 35.855 | 16.403 | 17.263 | 60.425 | 57.714 |
| 1–7 | 47–46 | Pre-MECO | 0.359 | 0.359 | 30.421 | 30.421 | 13.753 | 13.753 | 41.726 | 41.726 |

Estimates from pre-MECO, MECO (subgroups 1–3), and post-MECO.
*EWSD* Estimated Within-Sample Diversity.

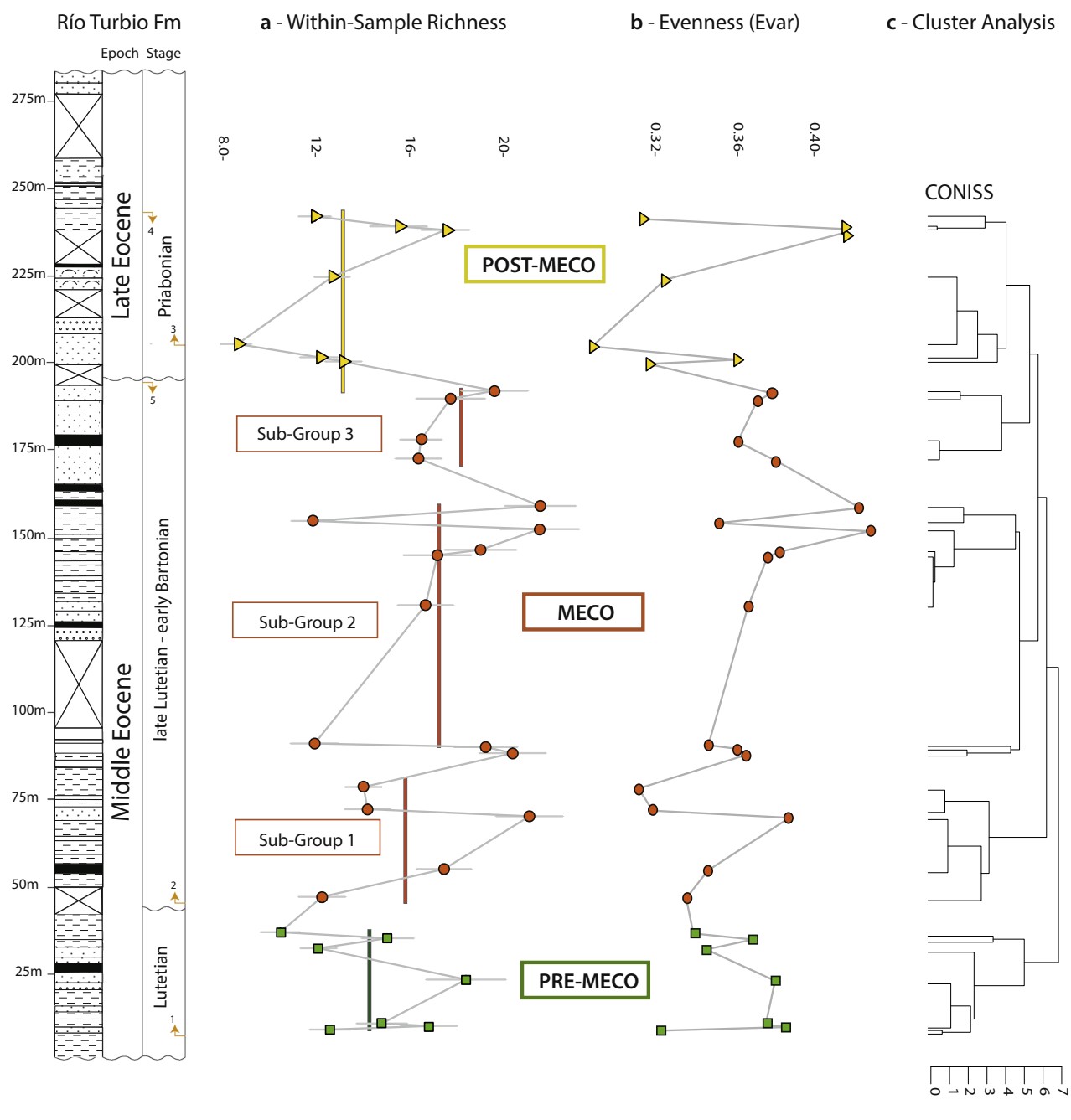

**Fig. 2 Floristic diversity across time in southern Patagonia. a** Within-sample richness (coverage level= 0.8) with bootstrapped 95% confidence intervals and **b** evenness for pre-MECO (green), MECO (dark-orange) with three subgroups and post-MECO (yellow) samples against the composite section of the Río Turbio Formation (see references in Supplementary Fig. 1). Vertical bars in **a** denote mean diversity within each major group of samples (or intervals) detected from the constrained cluster analysis (**c**). See Table 1 for further details. Arrows at the Stage column indicate major dinocyst events from the sampled composite section of the Río Turbio Formation; (1) Lowest Occurrence of *E. dictyostila*; (2) Lowest Common Occurrence of *E. dictyostila*; (3) Lowest Common Occurrence of *T. filosa*; (4) Highest Occurrence of *T. filosa*; (5) Highest Common Occurrence of *E. dictyostila* . See Supplementary Fig. 5 for further details.

ranging from samples 1–7 (ca. 47–46 Myr, pre-MECO); Interval (B) ranging from samples 8–25 (ca. 41–39Myr, MECO), typically characterized by the dominance of the species *E. dictyostila*, that increases as much as 95% at some of these samples (Supplementary Figs. 3, 4); and Interval (C) ranging from samples 26–32 (ca. 36–26 Myr, post-MECO). The frequency of distinct dinocyst biogeographic groups (i.e. endemics, cosmopolitans) preserved across the MECO shows a very close similarity with that reported from the South Tasman Rise[6,7], in Australia (see Supplementary Fig. 3 and Supplementary Note 2 for further details).

Among continental palynomorphs, we identified 117 spore and pollen species (Supplementary Data 2, Supplementary Note 3) represented by 2 bryophytes, 3 lycophytes, 25 ferns, 11 gymnosperms, and 76 angiosperms. We used these continental palynomorphs to empirically estimate biodiversity and explore major trends in vegetation across the three intervals detected based on the frequency of dinocyst species.

Our analyses on fossil spores and pollen grains indicate that our three intervals preserve relatively rich floras (Table 1; Figs. 2–3); adjusted for coverage (=0.8) we detect a ~25% increase

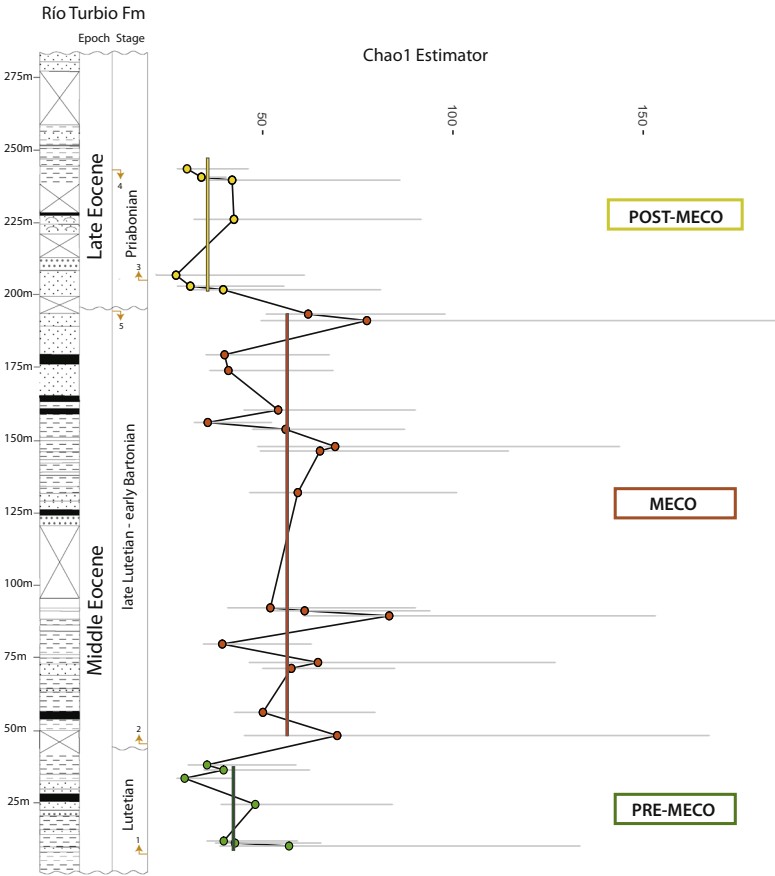

**Fig. 3 Floristic diversity across time in southern Patagonia.** Within-sample Chao1 estimated richness with bootstrapped 95% confidence intervals for pre-MECO (green), MECO (dark-orange) and post-MECO (yellow) samples from southern Patagonia the composite section of the Río Turbio Formation (see references in Supplementary Fig. 1). Vertical bars denote mean Chao1 diversity within each major group of samples (or intervals) detected from the continental constrained cluster analysis. Note that, in average, the within-sample Chao1 estimator is ~40% higher during the MECO than pre-MECO and post-MECO intervals. See Table 1 for further details. Arrows at the Stage column indicate major dinocyst events from the sampled composite section of the Río Turbio Formation; (1) Lowest Common Occurrence of *E. dictyostila*; (2) Highest Common Occurrence of *E. dictyostila*; (3) Lowest Common Occurrence of *T. filosa*; (4) Highest Occurrence of *T. filosa*; (5) Highest Common Occurrence of *E. dictyostila*. See Supplementary Fig. 5 for further details.

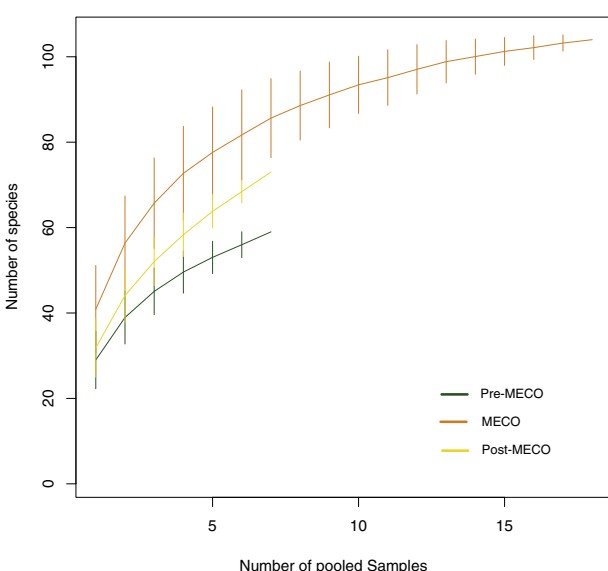

**Fig. 4 Species accumulation curves for pre-MECO, MECO, and post-MECO intervals.** Vertical lines denote bootstrapped 95% confidence intervals.

in within-sample richness from pre-MECO (interval A) to MECO (interval B) and a ~25% decrease from MECO to post-MECO (interval C). This perceived increase in richness can be, in part, due to an increase in evenness (Fig. 2); even samples tend to have a higher richness[8,9]. However, the richness estimator Chao1, which requires no assumptions about an underlying species abundance distribution[10], confirms both the increasing richness trend from pre-MECO to MECO (39%) and the decreasing trend from MECO to post-MECO (~38%) (Table 1, Fig. 3). Consistent with this, rarefaction curves for each interval indicates that MECO samples are relatively more diverse than pre-MECO and post-MECO samples (Fig. 4).

Furthermore, within the MECO (or interval B), we recognize three minor subgroups of samples (subgroup 1–3) with similar pollen and spore species according to our continental cluster analysis (Fig. 2c). Across these three subgroups of the MECO, we detect a clear inverse relationship between the abundance of ferns and angiosperms (Figs. 5, 6). At the beginning of the MECO (subgroup 1, samples 8–12), ferns highly increase in abundance (ca. 60%) with Cyatheaceae, Dicksoniaceae, and Osmundaceae as the most frequent families. At this peak of ferns, the abundance of angiosperms decreases dramatically (from 70 to 30%). At the core of the MECO (subgroup 2, samples 13–21), ferns drop to a minimum, while angiosperms become dominant (80%). Apart from the dominant lineages (i.e. southern beeches and

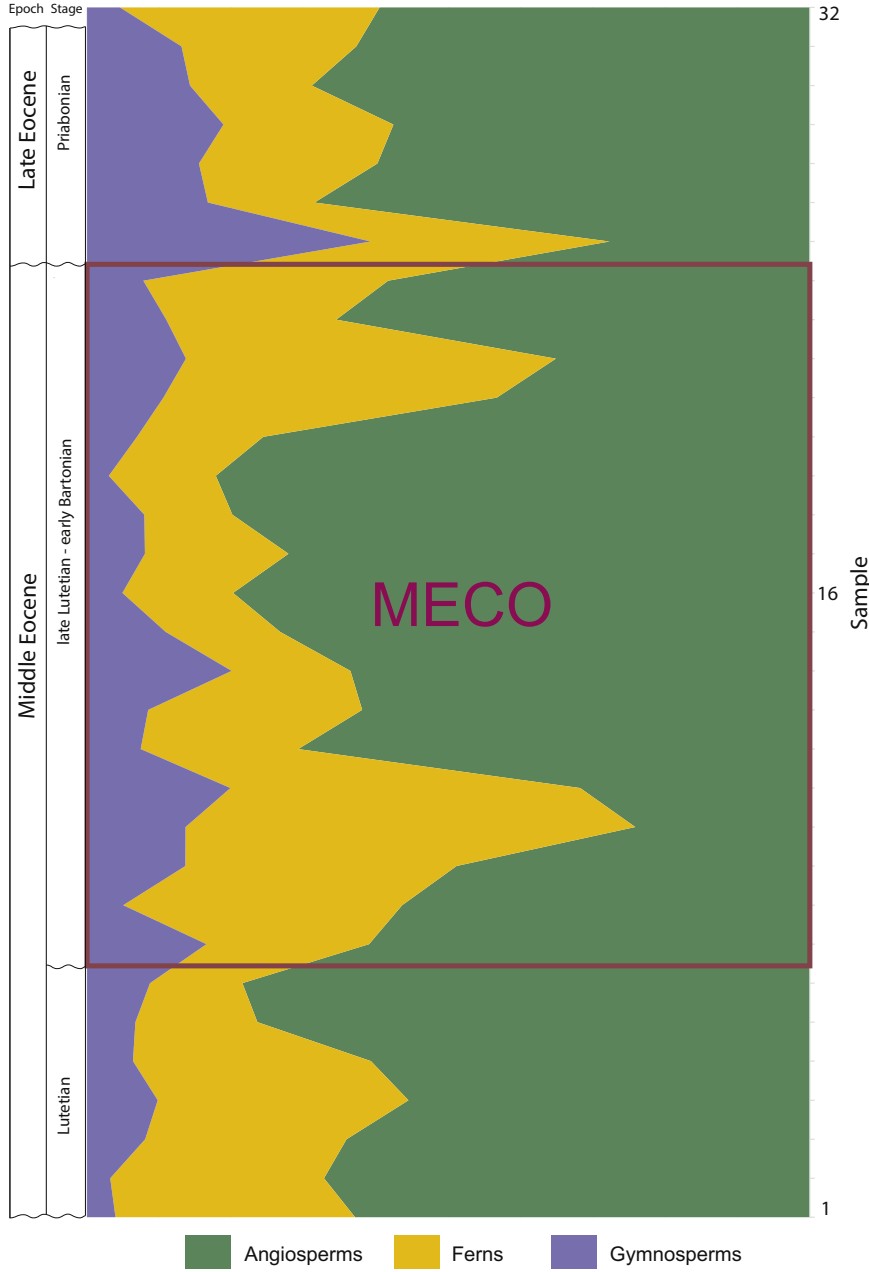

**Fig. 5 Relative frequency of the major plant groups through time.** Note the peak in abundance of ferns at the onset and end of the MECO.

podocarps), other gondwanan families (e.g. Myrtaceae and Proteaceae) became important elements (Fig. 6). At the end of the MECO (subgroup 3, samples 22–25) ferns rise again to maximum values (ca. 60%) while angiosperms decrease at the same time. Tropical taxa occur commonly across the entire sampled section, in particular palms, *Cupania*, *Ilex*, Malpighiaceae, Olacaceae, among others (Fig. 7; Supplementary Fig. 6; Supplementary Note 4). Some of them, however, are widely distributed throughout the MECO such as some angiosperms (e.g. *Ceiba*, *Cardiospermum*, *Trimenia*) and ferns (e.g. Anemiaceae, *Cnemidaria*, Schizaceae) (Supplementary Note 4).

Overall, pre-MECO and post-MECO samples not only show similar richness estimates (Table 1), but also contain comparable abundances of the major plant groups (Figs. 5, 6). Despite these similarities in diversity and abundance, the composition of the spore-pollen assemblages indicates a different scenario; distances among samples in ordination space (Fig. 8) support

dissimilarities, especially between pre-MECO versus MECO and post-MECO spore-pollen samples. Samples representing the pre-MECO plot to the right on the NMDS axis 1, and occupy a distinct region of the plot, showing they are compositionally distinct from post-MECO and MECO samples. These last two groups of samples, in contrast, slightly overlap, probably due to a gradual—rather than a sharp—compositional transition between them. Our constrained cluster analysis based on spore-pollen abundance also shows a comparable assembly, with pre-MECO samples sister to the MECO and post-MECO samples (Fig. 2c).

## Discussion

America's southernmost floras were impacted by the mid-Eocene greenhouse warming event. We found evidence to support that plant richness increased jointly with increasing world temperatures and atmospheric $CO_2$. Although richness estimates at the

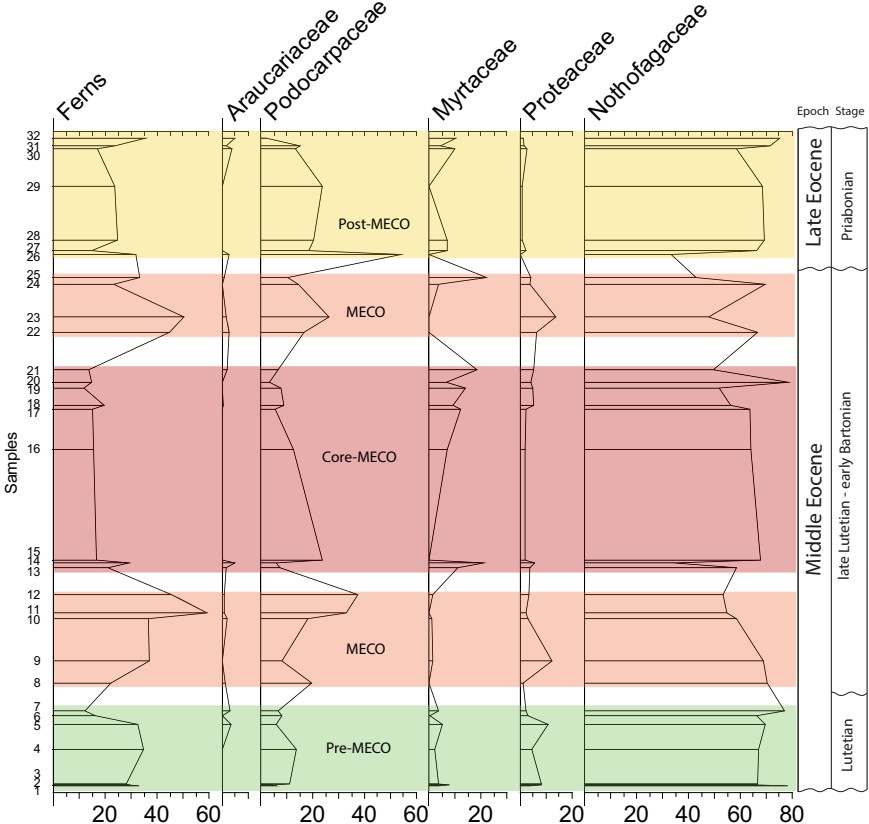

**Fig. 6 Relative frequency of the most common plant groups across the recognized intervals.** Note the increase in abundance of Proteaceae and Myrtaceae during the MECO.

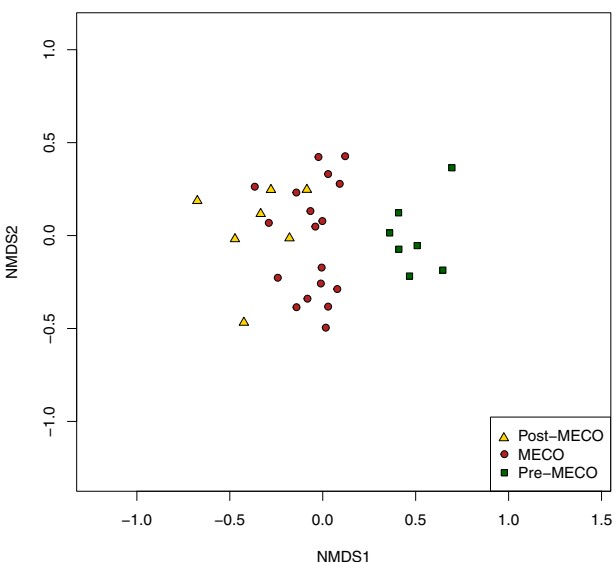

**Fig. 7 Distances among fossil-bearing samples in ordination space.** Our NMDS analysis supports dissimilarities, especially between pre-MECO against MECO and post-MECO spore-pollen samples. Pre-MECO samples plot to the right on axis 1, occupying a distinct region of the plot while MECO and post-MECO samples slightly overlap.

MECO from lower latitudes (Colombia) are considerably higher than those reported from Patagonia[11]; Tables 1, 2), the magnitude of this increase—from pre-MECO to MECO—is fairly similar (>35%) based on our non-parametric estimator. The enormous difference in plant richness between South American lowest

(~6°N) and highest (~51°S) latitudes (more than twofold) lead us to assume that the latitudinal diversity gradient (LDG) was already well established in South America by mid-Eocene times. This is before the onset of the oldest major glaciation in Antarctica close to the Eocene/Oligocene boundary (~34 Myr ago), a time of rapid global cooling and pronounced shift in the Earth's climate from greenhouse to icehouse[12]. The existence of this LDG during greenhouse conditions was also supported by other Eocene paleofloras[9] and marine invertebrates[13], suggesting that temperature may have not been the primary driver of the LDG through deep time[14]. Our high southern latitude paleofloras, even during the MECO, show patterns of relatively high dominance and low evenness; particularly Nothofagaceae, Podocarpaceae, Proteaceae and Myrtaceae tend to dominate the assemblages. Together, these Gondwanan lineages comprise ~30% of the total mid-Eocene diversity in southern Patagonia, and this figure rose during global cooling events up to 50%[15].

The equable climatic context of the mid-Eocene promoted the dispersal of tropical or subtropical taxa to the highest southern latitudes (Supplementary Fig. 6). For example, we documented four morphotypes assigned to palms (Arecaceae), and several other eudicot angiosperms (e.g. *Anacolosa*, Bombacoideae, *Cupania*, Malpighiaceae, Olacaceae, Sapindaceae) and ferns (e.g. *Cnemidaria*, Anemiaceae) that no longer occur in Patagonia; see Supplementary Note 4). Some of these taxa have been also documented on the basis of the megafloristic record from the Río Turbio Formation (e.g. refs. [16–18]). The subsequent cooling and aridification events of the Oligocene and, particularly the Miocene, pushed northwards these tropical-affinity taxa. Overall, our evidence demonstrates that the mid-Eocene greenhouse world favoured the penetration of neotropical migrant species to the highest latitudes; the combination of these neotropical migrants

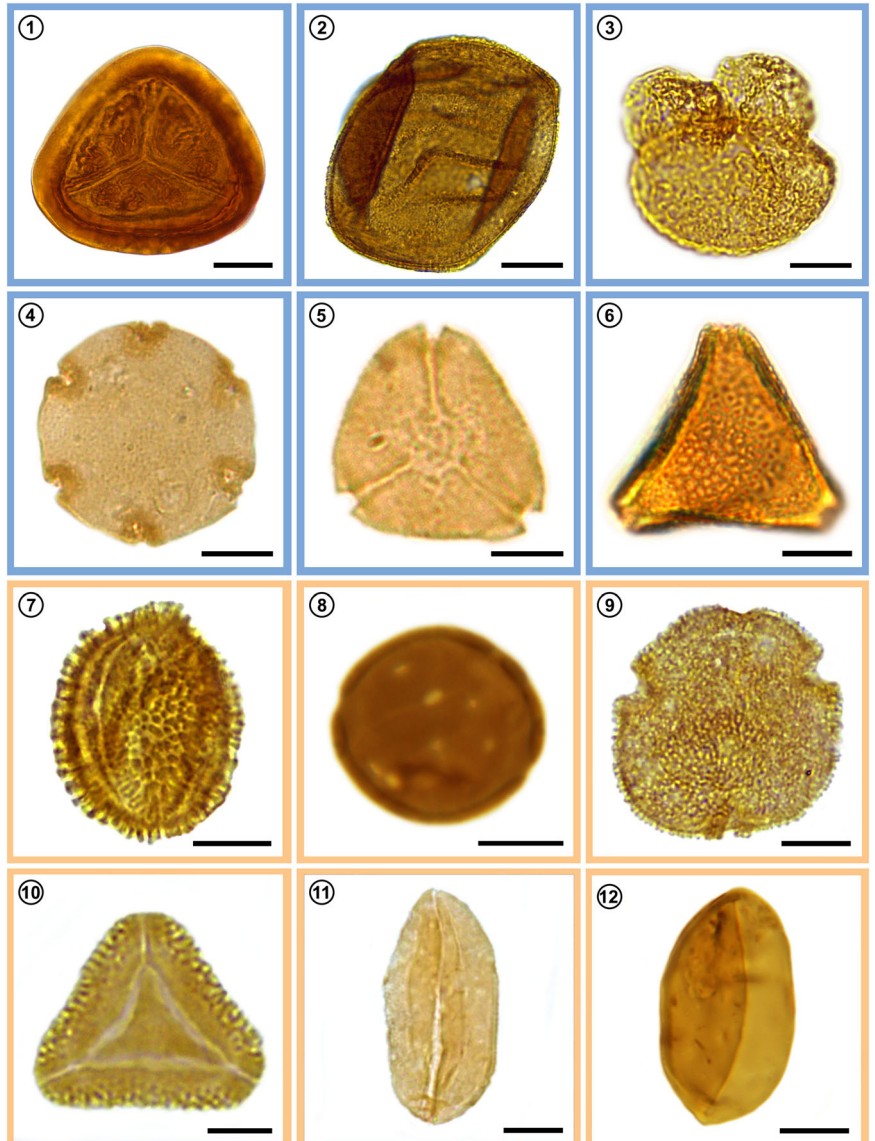

**Fig. 8 Selected spore-pollen species from the Eocene of southern South America.** (1–6) Gondwanic elements (blue square); (7–12) Tropical elements (orange square). (1) *Cyatheacidites annulatus*, sample 32 V18(1); (2) *Araucariacites australis*, sample 14 J44(2); (3) *Podocarpidtes elegans*, sample 28N41(4); (4) *Nothofagidites rocaensis*, sample 17 E40(1); (5) *Myrtaceidites verrucosus*, sample 14 H38(3); (6) *Propylipollis pseudomoides*, sample 19C22(1); (7) *Ilexpollenites anguloclavatus*, sample 13N57(4); (8) *Perisyncolporites pokornyi*, sample 3 I12(3); (9) *Bombacacidites isoreticulatus*, sample 16 Y39(2); (10) *Cupaneidites insulares*, sample 19 D13(4); (11) *Arecipites regio*, sample 3 R40(1); (12) *Psilamonocolpites medius*, sample 6 S31(1). Scale bar is 10 µm. Taxonomic names are followed by the slide number and England Finder coordinates.

**Table 2 Diversity estimates derived from spore-pollen records from Neotropics[11].**

| Age (Mya) | Interval | Evenness neotropics | EWSD (coverage = 0.8) | EWSD (Chao1) |
|---|---|---|---|---|
| 37–38 | Post-MECO | 0.536 | 42.483 | 74.782 |
| 38–41 | MECO | 0.658 | 47.457 | 118.149 |
| 42 | Pre-MECO | 0.519 | 37.573 | 102.674 |

Estimates from pre-MECO, MECO, and post-MECO.
*EWSD* Estimated Within-Sample Diversity.

along with the persistence of southern Gondwanan natives may have triggered the gradual increasing diversity that we observed across the MECO. Interestingly, our mid-Eocene peak in diversity mirrors the pattern seen in the South American mammal fauna,

which records the highest richness estimates for the Cenozoic from Patagonia during the Barrancan Mammal Age[19].

Terrestrial evidence on how rainfall patterns shifted over the MECO is sparse. Paleosol evidence from western North America and southern South America indicated that during this warming event, subhumid or semi-arid conditions prevailed, respectively[20,21]. Lithofacial and pollen records from Asia (northwestern China) indicate a rapid aridification step across the MECO[22]. Our analysis detects a strong reduction of humid-demanding taxa along with an increase of arid-tolerant angiosperms (e.g. Proteaceae and Myrtaceae) (Fig. 6), suggesting sub-humid conditions across the terrestrial zenith of the MECO (our core MECO). However, the onset and the end of the MECO are typically characterized by peaks in abundance of ferns (up to 60%), along with other wet-demanding taxa, indicating humid to hyper-humid conditions. Whether or not these shifts in humidity are linked to regional or global conditions remain to be tested.

Fossils revealing the floristic response to the MECO warming allowed us to estimate the magnitude of this increased in diversity in America's southernmost latitudes. In particular, we infer that greenhouse conditions promoted the diversification of austral floras, although plant richness in this region was remarkably lower than low-latitude counterparts at equivalent times. The subsequent Antarctic glaciation (early Oligocene) and widespread aridification (late Miocene) may have even accentuated such difference by gradually impoverishing Patagonian biotas. Overall, our study supports the notion that there has been a massive turnover from rich mid-Eocene rainforest biomes across the South American highest latitudes through the current steppe-dominated landscapes.

## Methods

**Fossiliferous localities**. Samples were collected from the shallow-marine Río Turbio Formation in southern Patagonia (Fig. 1). The Río Turbio Formation preserves a thick shallow-marine and estuarine succession characterized by sandstones, limestones, and conglomerates interbedded with clay horizons accumulated in coastal marine, wave- and tide-dominated shallow water environments[23,24]. Our high-resolution record encompasses the MECO as well as pre- and post-MECO floras. For the first time, we quantified plant species richness in the southernmost regions of South America during the globally warm mid-Eocene Epoch using palynological data (pollen and spores). Other fossils preserved in these deposits include terrestrial (leaves (e.g. ref. [25]), trunks (e.g. ref. [26])) and marine (shells (e.g. ref. [27]), foraminifera (e.g. ref. [28]), and dinocysts (e.g. ref. [29])) remains. The spore-pollen bearing sediments are constrained as the mid-late Eocene (~46–34 Myr) based on foraminifera[30] and dinoflagellate cyst (e.g. ref. [29]; this study (see Supplementary Note 2) data. Here, we studied pollen samples collected from a ca 400 m section spanning ~10 million years of the Eocene in order to better understand the effects of climatic change on continental biotas.

**Palynology**. A total of 53 samples from the Río Turbio Formation were processed. Palynomorph and dinocyst preparations were undertaken at the Museo Argentino de Ciencias Naturales and followed a basic procedure of maceration, chemical digestion of silicates (hydrofluoric acid), fluorosilicates (clorhidric acid), and a light oxidation to remove excess of amorphous matter (2 min in 70% nitric acid). Finally, residues were concentrated and mounted onto slides. Residues were sieved with 25 μm and 10 μm meshes. Fifty-one samples yielded abundant pollen, spores and dinoflagellate cysts. Slides were scanned under a transmitted light microscope Leica DM 500. Spores, pollen grains, and dinocysts were photographed by a Leica camera ICC50 HD. A mean of 354 spores and pollen grains and 285 dinocysts were counted per sample. Slides are housed at the Museo "Padre Jesús Molina" under the catalogue numbers 21647–21699, prefixed MPM-PB. We removed some of the samples (e.g. coal seam samples) for the subsequent biodiversity and abundance analyses, particularly those that we interpreted as having been deposited in a more continental paleoenvironment. Mudstones and fine sandstones preserving relatively high frequencies of dinocysts (representing temporary marine incursions) contain higher proportions of pollen from both wind-pollinated (e.g. podocarps and southern beeches) and insect-pollinated (e.g. malpighs, mallows, and palms) families, which may have originated a considerable distance inland from the coast. These samples therefore represent a much larger source area (regional to sub-continental), compared to the local signal contained in the coal measure samples, as previously tested[31].

**Quantitative analysis**. We conducted all analyses using the open-source software R[32]; see Supplementary Note 5 for R scripts. We arranged spore-pollen data from the Río Turbio formation in a $32 \times 117$ matrix with samples and taxa in which each cell contained count data for all taxa of the selected samples (Supplementary Data 2). We also included in our analysis a $21 \times 4375$ matrix (sedimentary section R1[11]) from the Neotropics encompassing the MECO (42.3–37.7 Mya) in order to compare floristic diversity estimates. We conducted two cluster analysis to explore sample associations (Q-mode) based on either marine (dinocysts) or continental (pollen and spores) palynomorphs. We used the 'chclust' function of R/rioja[33] that performs a constrained cluster analysis of a distance matrix, with clusters constrained by sample order. The distance matrix used was the Bray–Curtis metric[34] and the agglomeration method was the CONISS[33]. We ordinated the samples and species using NMDS (non-metric multidimensional scaling) using vegan R package[35]; NMDS is considered one of the most robust unconstrained ordination methods in community ecology[36]. We estimated $E_{var}$ evenness, recommended among other evenness indices[37]. For estimating biodiversity, we standardized samples to equal levels of size and completeness, or 'coverage' of species[38], also known as shareholder quorum subsampling, or coverage-based rarefaction. Standardizing sampling by coverage rather than sample size has proven to be a more powerful and less biased approach to estimate richness[38]. We estimated expected richness within samples based on our abundance matrix using iNEXT R package[39]

and among samples from the pre-MECO, MECO, and post-MECO intervals using vegan R package[35]. For within-sample richness, we calculated the expected richness at a coverage level of 0.8, enough to include most of the samples of our dataset. We also estimated Chao1 richness estimator, which uses singletons (species represented in the sample by only one individual) and doubletons (species represented in the sample by exactly two individuals) to estimate the number of unobserved species. This non-parametric estimator has a more rigorous framework of sampling theory than parametric estimators or curve extrapolations[40]. The bootstrapped 95% lower and upper confidence limits are also presented for all richness estimators.

**Reporting summary**. Further information on research design is available in the Nature Research Reporting Summary linked to this article.

## Data availability

Slides are housed at the Museo "Padre Jesús Molina" under the catalogue numbers 21647–21699, prefixed MPM-PB. The authors declare that the data that support the findings of this study are available within this paper and its Supplementary Information files, and are available from the corresponding author on reasonable request.

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

## Acknowledgements

We thank O. Cardenas and S. Mirabelli for assistance with palynological processing; A. González for assistance with drawing; V. Guler for assistance with stratigraphy; M.S. Candel for assistance with dinocyst taxonomy; A. Yañez and J.N. Viera Barreto for assistance with phytogeography. One anonymous reviewer, Carina Hoorn, and Alina I. Iakovleva improved enormously our first version of the manuscript. This work was partially supported by Consejo Nacional de Investigaciones Científicas y Técnicas (PIP 2014–0259) and Agencia Nacional de Investigaciones Científicas y Técnicas (PICT 2017–0671).

## Author contributions

D.A.F., L.P., and V.D.B. contributed with the measurement and description of the stratigraphic sections and collection of samples for palynological analyses. D.A.F. performed the palynological counting (spore-pollen and dinocysts) and palynological assemblage description, captured palynomorphs images and designed the figures illustrating them. D.A.F., L.P., and V.D.B. wrote all draft and revised manuscript versions, which includes original and revised text supplied by co-authors. L.P. contributed with quantitative analysis and illustrations. M.S.G.E. contributed with the dinocyst and stratigraphic analysis and illustrations. D.A.F., M.C.T., and V.D.B. contributed with the analysis of spore-pollen botanical affinity and phytogeography. All authors read and contributed to the manuscript.

## Competing interests

The authors declare no competing interests.
