## [Peer Review File · Communications Biology]

Reviewers' comments:

Reviewer #1 (Remarks to the Author):

The paper by Fernández et al focuses on palynofloras from the MECO of southern Patagonia, and finds differences in diversity and composition compared to the pre- and post- MECO phases. Overall the paper is well-written, the dataset is a valuable one (as the authors note, very little is known about the MECO in the terrestrial realm), and the authors take on some interesting questions that would be relevant to people from a range of disciplines beyond palynology. I do have some concerns however which I think need addressing before this paper can be published. I've started off with the main, overarching points, and then more specific, minor issues follow:

Main issues

1. Most of the samples are from the MECO interval, and there's very little context above and below the event. It's therefore very hard to know just how remarkable this event is: how high is the diversity relative to background conditions, given that you only have four samples before the onset of the MECO? And how atypical is it to have these tropical elements present in the flora? Supplementary Figure 5 suggests that there were at least some tropical taxa beforehand, with a similar abundance to some of the MECO samples (especially those in subgroup 2, in the core of the MECO). Without some knowledge of the diversity and composition above and below this section I'm not convinced that the authors can properly contextualise their data, and really assess the impact of the MECO warming.

2. I'm not entirely convinced by your choice of diversity measures. Standing/range through diversity always suffers from the Signor-Lipps effect, with taxon first and last appearances never representing true first and last appearances but occurring relatively later or earlier respectively. This means that the taxon ranges get bunched up in the middle of the time series, even if they extend all the way to the ends, making diversity appear artificially higher in the middle and lower at the edges. It's an even bigger problem with this sort of dataset that covers a relatively short time period, and where the taxa probably have their first and last occurrences below and above the studied section (it's not clear from the methods if the authors did anything to correct for this, such as extending the ranges of those taxa that are known to occur earlier or later in time all the way to the ends of the section before tabulating diversity). In your case it will mean that MECO diversity will be biased towards higher values, as shown in Figure 2.

I'm also not sure I follow the within-flora diversity aspect: not only does it seem quite hard to interpret in any intuitive way (perhaps it would make more sense in a spatial analysis when the interest is in linking communities to a regional species pool or something?), I'm also not clear why the values in Table 1 are all in the 30% while in Figure 3 they are 0.6 – 0.9. I also note that when Figure 3 is considered, phase B has a much higher within-flora diversity than phases A and C, but when it is broken down in Table 1 into 3 subgroups the difference between the phases becomes much less pronounced – is this an issue with having more samples in phase B versus A and C, which then gets evened out when the dataset is split into the 5 phases in Table 1? I think my fundamental issue is that I really don't understand what this metric is telling me, and it certainly doesn't correspond to 'Raw species richness' (line 199), at least as I understand the term. Both this and the range through richness also make no corrections for differences in sample size.

Rarefied richness makes a lot more sense, and as with within-flora diversity the difference between the MECO and the pre-and post- phases is much less pronounced in Table 1 than the range through richness plot suggests in Figure 2, which again suggests that the impact of the MECO is overstated in the paper. I also encourage the authors to look at coverage-based rarefaction (SQS in John Alroy terms; see Chao and Jost 2012) because coverage is a better measure of sampling completeness than count size. It would also be useful to look at evenness, because changes in evenness through a section will give the appearance of changes in richness

when sampling is incomplete (relatively more even samples will appear to have higher richness), and possibly richness estimators as well because these appear to be a bit less affected by this evenness issue.

My suggestion therefore is to alter how the diversity data are measured and presented in the paper. Rather than showing just range through richness at the edge of a figure which is dominated by dinoflagellate data (Figure 2), I would have a dedicated figure showing within-sample rarefied richness against height in the section (with confidence intervals to show the error in the richness estimates), plus a suitable evenness metric and within-sample extrapolated richness (e.g. the Chao1 estimator) as well. The sporomorph cluster dendrogram that is currently relegated to the supplementary figures could also occur here. It might also be worth thinking about looking at ordinations, such as NMDS or correspondence analysis. If the MECO interval really is different in its composition, then this would be picked up more clearly than in the constrained cluster analysis (the PETM analysis in Wing et al 2005 gives a nice example of this kind of analysis).

If you want to show within-phase richness to compare pre-MECO, MECO and post-MECO diversity, sample based rarefaction curves sensu Gotelli and Colwell (2001) would be a better way to show this than the box plot in Figure 3, and can be carried out either with sample size or coverage-based rarefaction (see Chao and Jost 2012). Species richness estimators can also be calculated and plotted in this way, e.g. the Chao2 estimator for comparing richness among groups of samples.

There are lots of examples of these sorts of analyses, applied to sporomorph data, in the literature, see for example Harrington and Jaramillo (2007), Harrington et al (2004), Jaramillo et al (2010), Jardine et al (2018), Wing et al (2005), plus various other papers by the same authors.

I've written a lot here, but since the main thrust of your paper rests on the atypical nature of the MECO in terms of plant diversity and composition, you need to make sure you use appropriate analytical approaches so that these claims can be supported.

3. As it stands the paper is well under the limit for word count and number of display items that the journal allows, and I think it could really benefit from a bit more detail and careful discussion, rather than going straight for a headline story that isn't fully supported (at least with the way the data are presented now). As noted above, more data figures would probably help here. In addition to the analyses I've mentioned, bringing in the sporomorph relative abundance plots from the supplementary information into the main paper would probably be useful, and if anything needs cutting to allow for this, I think the dinoflagellate data in Figure 2 and the sporomorph images in Figure 4 could go into the supplementary information instead.

4. I appreciate the inclusion of the sporomorph count data as a supplementary file, but it would be useful to have similar data for the dinoflagellates, rather than these just being listed by name in Supplementary Table 1. The R code you used to analyse your data would also be valuable, so that everything is fully reproducible.

Minor points

Lines 26-28: I understand the value of paleo-records for forecasting Earth's future, but I'm not convinced this does provide a good analogue for near-future conditions: so much of your reconstruction depends on taxa shared with a vegetated Antarctic (or the Gondwanan continents more generally), and out of the tropics migration routes that probably wouldn't be available in today's fragmented habitats. I suspect the lessons from a 40 million year old record provide limited data for forming solid predictions about Earth's immediate future, and I suggest toning down this sentence a bit (impressive though it sounds).

Line 43: The floras in reference 5 (Wilf et al 2005) are early and middle Eocene but not PETM.

Lines 138-141: does this imply relatively wetter conditions at the onset and end of the MECO, as defined by subgroups A and C when ferns increase in abundance?

Line 146: I'm really not convinced by this comparison with extant floras. Holocene pollen samples are based around a different taxonomic concept to deeper time material (extant genera and families versus morphologically-circumscribed form taxonomy) and so I don't think the two things are really comparable. I realise that you are following Jaramillo et al here, but I was sceptical of this in their paper too. I think this statement, and the one that follows on comparing Patagonian richness through time, really needs unpacking and justifying, with the rarefied richness estimates given in each case (graphically if you like) and the rationale for this comparison given.

Line 149: And what are these major implications? If you want to say this then I think it needs at least a few sentences specifying what these are (this comes back to my earlier comment about the final sentence in the abstract).

Lines 182-190: this isn't vital here, but for future work it would be interesting to look at the coal samples in a bit more detail, to give more information on the local (swamp?) taxa in comparison to the regional vegetation. Getting whatever information you can from the cuticles (mentioned in supplementary note 3) would also be worthwhile.

Supplementary figures 4 and 5 - are the taxon abundances as specimen counts? These would make more sense as percentages.

References

Chao, A. & Jost, L. (2012) Coverage-based rarefaction and extrapolation: standardizing samples by completeness rather than size. *Ecology*, 93, 2533-2547.

Gotelli, N.J. & Colwell, R.K. (2001) Quantifying biodiversity: procedures and pitfalls in the measurement and comparison of species richness. *Ecology Letters*, 4, 379-391.

Harrington, G.J. & Jaramillo, C.A. (2007) Paratropical floral extinction in the Late Palaeocene-Early Eocene. *Journal of the Geological Society, London*, 164, 323-332.

Harrington, G.J., Kemp, S.J. & Koch, P.L. (2004) Palaeocene-Eocene paratropical floral change in North America: responses to climate change and plant immigration. *Journal of the Geological Society, London*, 161, 173-184.

Jaramillo, C.A., Ochoa, D., Contreras, L., Pagani, M., Carvajal -Ortiz, H., Pratt, L.M., . . . Vervoort, J. (2010) Effects of rapid global warming at the Paleocene-Eocene boundary on Neotropical vegetation. *Science*, 330, 957-961.

Jardine, P.E., Harrington, G.J., Sessa, J.A. & Dašková, J. (2018) Drivers and constraints on floral latitudinal diversification gradients. *Journal of Biogeography*, 45, 1408-1419.

Wing, S.L., Harrington, G.J., Smith, F.A., Bloch, J.I., Boyer, D.M. & Freeman, K.H. (2005) Transient floral change and rapid global warming at the Paleocene-Eocene boundary. *Science*, 310, 993-996.

Reviewer #2 (Remarks to the Author):

The Eocene of Patagonia is one of the most fascinating research topics, when studying the effects of climate change on vegetation. At the time South America was still connected to Antarctica, with temperatures fluctuating but also reaching the highest values recorded in the Cenozoic. There is a broad body of literature on the diverse Eocene Patagonian flora, in which the 'mixed' nature, Gondwanan/Antarctic and tropical composition, is highlighted (e.g. Wilf et al., 2003, 2005, Barreda & Palazzesi, 2007; Vento & Pramparo, 2018). Particularly Vento & Pramparo (and references therein) give a detailed account of the mixed character of the Patagonian Eocene flora as recorded in the Rio Turbio Fm and they also point at the importance of the MECO. What is less clear is how the transition of this flora into cooler climate is expressed.

In their paper Fernandez et al. document document the Patagonian Rio Turbio Formation and provide a detailed log and new high resolution dating based on a dinoflagellate analysis. They also provide a quantitative sporomorph analysis that assesses changes in pollen composition and diversity, and a statistical analysis that follows the biotic response before, at, and after the MECO. The paper is very interesting, well written and includes clear synthetic figures. The real selling point of the paper is the quantification of the biotic changes across the important MECO climate transition in this low latitude location.

Having said all this, there are some important issues that need to be addressed before the paper can be considered for publication.

1- The identification of the 'tropical' taxa. Tropical is used here as one of the hooks of the paper and uses the presence of Bombacaceae, Arecaceae, Malpigiaceae, among others, as key argument. However, the light microscopy photos of these taxa are not very convincing. The photo plate does not have a good resolution (maybe I have the low-resolution version?) and none of the 'tropical' taxa seem really tropical; the identifications are in some (critical) cases doubtful. The two Arecaceae (Fig. 4: photos 11 and 12), both could easily be fern spores, in particular the taxon identified as *Monosulcites perspinosus*. Nevertheless, *Arecipites minutiscabratus* also does look spore-like. Better images and identifications are necessary due to the important role that these taxa play in the paper!

Similarly, the taxon identified as *Bombacidites isoreticulatus* (photo 9), easily could represent a *Ceiba* (*speciosa* type?), a Bombaceae that tolerates temperate climate and is present today in Argentina. In addition, the taxon labeled as *Perisyncolporites pokorny* (photo 8) does not have the characteristics of this fossil. I am aware that this species, has been previously reported in the Rio Turbio Fm by the first author (in a different paper), so a more convincing photo should be included. In addition, the biogeography of *Cupania* and *Ilex* is not restricted to the tropics. It should be noted that it does not require a tropical climate to maintain some of the above taxa in Argentine. Did they adapt to a cooler climate after the Eocene? Maybe, but this should be discussed. Note for instance the paper on fossil macroremains of palm material in the Eocene by Romero in Ameghiana (1968) on *Palmoxydon patagonicum*, who says the following: "it is concluded that *P. patagonicum* has intermediate characters with the subfamilies Sabaloideae, Coccoideae and Bactrioideae, which are now represented in Chile and Argentina. On this ground it is suggested that *P. patagonicum* might belong to the ancestral stock from which these subfamilies evolved".

In the abstract (line 23) therefore, do not use "very close to that occurring today in the Neotropics". Although at the time there is a connection/exchange with the tropics, that statement does not seem appropriate for the taxa that are listed.

Lines 110-112: the fern taxa that are listed are mountain ferns in the lower latitudes, some considered immigrants of southern origin in today's northern South American mountains. They are not genera typical of the tropical lowlands.

2 - I miss a reflection on what already is known from this formation and how this new study contributes to existing knowledge. For instance, Guerstein et al. is referred to for the taxonomic composition. When comparing the dinoflagellate diagram in the Guerstein paper (which includes

the Rio Turbio) with the current paper, it is virtually identical. How does present study add to this?

Methods: lines 185-190 the sampling rationale is clear. The authors highlight that mudstones were selected to avoid local bias. However, is it not the case that mudstones will have bias toward wind-pollinated taxa? They may represent a larger area, but not necessarily the diversity of that area.

Figures: supplement the log in figure 1 should have the m-scale next to the first section so that we can see the thickness in meters right away. The tiny scale in the legend, and thickness of selected strata is not very helpful.

Carina Hoorn, 22nd of July 2020

Reviewer #3 (Remarks to the Author):

After maximum temperatures in the early Eocene, ensuing mid-late Eocene long-term gradual cooling ultimately lead to major expansion of Antarctic ice-sheets at the Eocene-Oligocene transition. This transitional period of cooling is characterized by a short-lived warming event – the Middle Eocene Climatic Optimum (MECO) that has been discovered in ocean drilling cores from the Atlantic, Pacific and Southern Oceans and in land-based marine sections in Italy and UK. The MECO is marked by a rapid negative shift in both oxygen and carbon isotope ratios and thought to reflect an increase in sea surface and bottom water temperatures by up 5-6°C. However, well-dated high-resolution records encompassing the MECO are still lacking in terrestrial realm and it is still largely unknown how the MECO-event affected the vegetation dynamics (i.e. diversity increase, plant migration patterns etc.).

From this point of view, the present manuscript of Fernandez et al. entitled "Impact of mid Eocene greenhouse warming on America's southernmost floras" is an important and very interesting paper dealing with the terrestrial paleoecological repercussions of the MECO in southernmost South America. Based on new quantitative (including cluster analysis) palynological results from a number of sections from Patagonia (famous Rio Turbio Formation), this publication represents a first evidence for extraordinary plant diversity during the MECO as a response to this greenhouse event in high austral latitudes. According to author's data, three intervals of different flora's diversity have been recognized at the beginning, within and at the end of MECO, suggesting a massive turnover from extraordinary diverse mid-Eocene rain-forest biomes through the modern steppe-dominated landscapes.

Some drawback of the article could be called the lack of isotopic analyses confirming the MECO event directly in the Rio Turbio Formation. Nevertheless, it is known that in Southern Hemisphere the dinoflagellate cyst assemblages, calibrated by isotopic data to the MECO, are very specific and are characterized by the unique in the Eocene record dominance of endemic species *Enneadocysta dictyostila*. Consequently, taking into account a well-defined calibration of sporomorph assemblages from the marine sediments of the Rio Turbio Fm to the lowermost common occurrences of *Enneadocysta dictyostila*, this can serve as a basis for the assertion of the MECO event.

I think that the paper of Fernandez et al. merits to be published in "Communications Biology". The presented results are novel and will be definitely of great interest to a large community of paleontologists, geologists and paleoclimatologists working on the Cenozoic. Statistical analysis is appropriate; conclusions are clear and convincing.

I have made a number of suggestions directly on the manuscript and supplementary information (pdf-file attached).

Here I just would like to make two comments of the Figures:

(1) Supplementary Figures 1 and 3:

The general stratigraphic scheme (Stages) and Dinocyst zones should be moved on the left of figure, while the Phases (I suggest to replace them in the text by Intervals) can be left on right.

(2) Supplementary Figure 6: Please precise which column demonstrates the present study.

I hope that my review will help.

Sincerely,

Alina I. Iakovleva

Geological Institute, Russian Academy of Sciences, Moscow, Russia.

**Impact of mid Eocene greenhouse warming on America's southernmost floras**

Damián A. Fernández^{1,5}, Luis Palazzesi^{2,5*}, M. Sol González Estebenet^{3,5}, M. Cristina
Tellería^{4,5}, Viviana D. Barreda^{2,5}

1 Laboratorio de Geomorfología y Cuaternario, CADIC, Bernardo Houssay 200, V9410,
Ushuaia, Argentina. fdamianandres@cadic-conicet.gob.ar

2 Sección Paleopalínología, Museo Argentino de Ciencias Naturales “Bernardino Rivadavia”,
Av. Ángel Gallardo 470, C1405DJR, CABA, Argentina. lpalazzesi@macn.gov.ar (*);
vbarreda@macn.gov.ar

3 Instituto Geológico del Sur (INGEOSUR), Universidad Nacional del Sur, Departamento de
Geología, San Juan 670, B8000ICN, Bahía Blanca, Argentina.
sol.gonzalezestebenet@uns.edu.ar

4 Laboratorio de Sistemática y Biología Evolutiva, Museo de La Plata, Paseo del Bosque
13 s/nº, B1900FWA, La Plata, Argentina. mariatelleria@fcnym.unlp.edu.ar

5 Consejo Nacional de Investigaciones Científicas y Técnicas (CONICET), Buenos Aires,
Argentina.

A major climate shift took place about 40 Mya ago —the Middle Eocene Climatic Optimum
~~of~~ MECO~~—~~ triggered by a significant rise of atmospheric CO₂ concentrations. The biotic
response to this MECO is virtually unknown in the terrestrial realm. Here, we reconstruct the
floras from America’s southernmost latitudes based on the analysis of terrestrially derived
spore and pollen from the mid-late Eocene (42-33.5 Mya) of southern Patagonia. We present
evidence for extraordinary plant diversity during the MECO, very close to that occurring
today in the neotropics. The combination of immigrants from the tropics and Antarctica,
along with the persistence of natives, favoured the increasing diversity observed during the
MECO. Our reconstructed biota reflects a greenhouse world and offers a climatic and
ecological deep time analog of an ice-free sub-Antarctic realm that may be our best means to
predict how a near future can be in the world's highest latitudes.

**Keywords:** ~~Fossils~~, MECO, ~~Greenhouse~~, Floras, Patagonia

The Earth has undergone a general cooling trend for the past ~50 Myr, culminating in
a continental-scale glaciation of Antarctica at the Eocene–Oligocene boundary. The Middle
Eocene Climatic Optimum (or MECO) occurred about 40 million years ago, interrupting that
cooling trend when vast amounts of CO₂ were injected into the atmosphere, and sea surface
temperature increased as much as 6°C (1). This warming event, widely recognized by a
prominent perturbation in both oxygen and carbon stable isotopes, lasted about 500–600
Kyr (2, 3).

Ecological models can potentially predict the impact of species diversity to rising
temperatures and atmospheric CO₂. However, only the fossil record provides empirical
evidence on how biodiversity is affected by long-term climatic transitions, even during global
warming events. For example, Cenozoic floras peaked in diversity during the widely explored
Paleocene-Eocene Thermal Maximum (or PETM) either at low (e.g. 4) or high (5) paleo-
latitudes of the American continent. The MECO may have also influenced terrestrial biotas,
yet the magnitude of this response remains largely unknown as most published data have
traditionally focused on the marine realm. For example, it is still unclear whether biotic
diversity increased, whether turnovers were gradual or step-like or whether tropical
immigrants were frequent at the highest latitudes during the MECO.

Here, we quantitatively estimate shifts in floristic diversity on the basis of the analysis
of terrestrially derived spore-pollen assemblages preserved in well-constrained marine
Patagonian deposits (Río Turbio Formation) encompassing the MECO as well as the pre- and
post-MECO. We used the dinocyst (i.e. dinoflagellate cyst) record to constrain the age of the
spore and pollen bearing sediments. Our study represents a new explicit picture of how floras
responded to a greenhouse event in America's highest austral latitudes and predicts how a
near future can be if carbon release continues to rise.

**Results**

We recovered well preserved palynomorphs (i.e. dinocysts, spores and pollen grains) in 53
samples of the Río Turbio Formation, southern South America (Fig. 1; Supplementary Fig.
1). From those, we selected 29 samples based on lithology (e.g. coal seams were removed
from the analysis) and paleoenvironmental settings (see Methods; Supplementary Note 1).
The dinocyst species are given in Supplementary Table 1 and Supplementary Fig. 2. We
detected three major phases of dinocysts based on our cluster analysis (Fig. 2), probably
driven by shifts in the frequency of *Enneadocysta dictyostila* through the section, which is the
key species of the Middle Eocene Climatic Optimum (MECO) in the southernmost latitudes
(see Supplementary Note 2). These three phases are: ~~Phase A) ranging from samples 1 to 4~~
(*ca.* 42 Myr, pre-MECO); ~~Phase B) ranging from samples 5 to 22~~ (*ca.* 40–39 Myr, MECO),
typically characterized by the dominance of ~~the~~ species *E. dictyostila*, that increases ~~as much~~
~~as 95% at some of these samples~~ (Fig. 2); and ~~Phase C) ranging from samples 23 to 29~~ (*ca.*
39–33.5 Myr, post-MECO). The frequency of distinct dinocyst biogeographic groups (i.e.
endemics, cosmopolitans) preserved across the MECO shows a very close similarity with that
reported from the South Tasman Rise (6,7), in Australia (see Supplementary Fig. 3 and
Supplementary Note 2 for further details).

Among the continental palynomorphs, we identified 118 spore and pollen species
(Supplementary Table 2) represented by 2 bryophytes, 3 lycophytes, 25 ferns, 11
gymnosperms and 77 angiosperms. We used these continental palynomorphs to empirically
estimate biodiversity and explore major trends in vegetation across the three phases detected
based on dinocyst species. We apply both a raw empirical approach (within flora diversity,
standing diversity) and a sampled-based estimation (rarefaction) to reconstruct
palaeodiversity on the basis of the spore and pollen record.

Our analyses indicate that our three phases contain diverse floras; within-flora
diversity estimates between phases A and C are not significantly different from each other

(Table 1; Fig. 3). The within-flora diversity from Phase B, is significantly higher than that
from Phases A and C (Table 1; Fig. 3). Adjusted for sample size (rarefaction) and standing
diversity estimations also indicate that Phase B contains the most diverse samples (Table 1).
Interestingly, this Phase B represents the MECO event. Within Phase B, we recognize three
minor sub-groups of samples (sub-group 1 to 3) with similar ~~pollen and spore~~ species
according to our continental cluster analysis (Supplementary Fig. 4). Across these three
subgroups of the MECO, we detect a clear inverse relationship between the abundance of
ferns and angiosperms (Supplementary Fig. 4). At the beginning of the MECO (sub-group 1,
samples 5–9), ferns highly increase in abundance (ca. 60%) with Cyatheaceae,
Dicksoniaceae, and Osmundaceae as the most frequent families. At this peak of ferns, the
abundance of angiosperms decreased dramatically (from 70% to 30%). At the core of the
MECO (sub-group 2, samples 10–18), ferns drop to a minimum, while angiosperms become
dominant (80%). Apart from the dominant lineages (i.e. southern beeches and podocarps),
other gondwanan families (e.g. Myrtaceae and Proteaceae) became important elements
(Supplementary Fig. 5). Tropical lineages also became common across the MECO, in
particular Areaceae, Aquifoliaceae, Malvaceae Bombacoideae, Sapindaceae, Olacaceae and
Malpighiaceae. At the end of the MECO (sub-group 3, samples 19–22) ferns rise again to
maximum values (ca. 60%) while angiosperms decrease at the same time.

Our Phase A (pre-MECO) and Phase C (post-MECO) not only show similar diversity
estimates (Table 1), but also comparable abundance values; they are dominated by
angiosperms followed by ferns; gymnosperms are slightly more abundant in Phase C
(Supplementary Fig. 4).

**Discussion**

America's southernmost floras were deeply impacted by the mid Eocene greenhouse
warming event. We found evidence to support that tropical lineages expanded to the highest
latitudes during this hyperthermal event; for example, we discovered five morphotypes
assigned to palms (Arecaceae), and several other eudicot angiosperms (e.g. Malpighiaceae,
Sapindaceae *Cupania*, Olacaceae *Anacolosa*, Malvaceae Bombacoideae) and ferns (e.g.
Cyatheaceae *Cyathea*, *Hemitelia/Cnemidaria*, Dicksoniaceae *Lophosoria*, Lygodiaceae
*Lygodium*) that became locally extinct from Patagonia, but occur today in lower latitudes.
Apart from these neotropical or temperate elements, we also found the typically recognized
southern gondwanan lineages —from which many of them no longer occur today in the
region— especially podocarps (*Dacrycarpus*, *Dacrydium*, *Lagarostrobos*, *Microcachrys*),
proteas (*Beauprea*, *Bleasdalea/Hicksbeachia*, *Embothrium*, *Lomatia/Gevuina*,
*Xylomelum/Lambertia*), and southern beeches (*Nothofagus*) (Fig. 4). Overall, our evidence
demonstrates that the mid Eocene greenhouse world favoured the penetration of neotropical
migrant species to the highest latitudes. The combination of these neotropical migrants along
with the persistence of southern Gondwanan natives may have triggered the gradual
increasing diversity that we observed across the MECO. Interestingly, our mid Eocene peak
in diversity mirrors the pattern seen in the South American mammal fauna, which records the
highest richness estimates for the Cenozoic from Patagonia during the Barrancan Mammal
Age (8).

An earlier peak in biodiversity occurred in the American continent during the
Paleocene-Eocene Thermal Maximum (PETM). Our data indicate that the response of the
southernmost floras to the MECO was comparable to that of the PETM. In particular,
adjusted for 208 specimens, we found 39 species while Jaramillo et al. (9) obtained 44 for the
PETM of the South American tropics, and adjusted for 171 specimens, we found 35.6 species
while Harrington & Jaramillo (10) obtained 35 species for the North American tropics. These

estimations lead us to believe that a high diversity pattern existed across the Paleogene
American floras during hyperthermal events rather than the strong latitudinal gradient that
prevails during icehouse periods of earth as today.

Terrestrial evidence on how rainfall patterns shifted over the MECO is sparse.
Paleosol evidence from western North America and southern South America indicated that
during this warming event, sub-humid or semi-arid conditions prevailed, respectively (11,
12). Lithofacial and pollen records from Asia (north-western China) indicate a rapid
aridification step across the MECO (13). Our analysis detects a strong reduction of humid-
demanding taxa (in particular ferns) along with an increase of arid-tolerant angiosperms (e.g.
Proteaceae, Myrtaceae) (Supplementary Fig. 5), suggesting sub-humid conditions across the
terrestrial zenith of the MECO (our sub-group B).

Overall, our study ~~demonstrates that there has been~~ a massive turnover from
extraordinarily diverse mid-Eocene rainforest biomes across the South American highest
latitudes through the current steppe-dominated landscapes. In particular, we infer that our
fossil plant assemblages were as rich as those recorded in recent times at the neotropics based
on adjusted for sample-size estimations (9), yet the subsequent Antarctic glaciation and
widespread aridification caused a major diversity loss; we estimate that about 75% of the
species became extinct if we compare past (our data) and present (14) rarefied diversity in
southern Patagonia. Our estimates may have major implications for projecting future effects
of increasing CO₂ on high latitude plant biodiversity.

Fossils revealing this dramatic floral response to MECO warming allowed us to
estimate the magnitude of this diversity event in America's southernmost latitudes. The lack
of further diversity studies across this transient event prevents addressing whether or not
these short-term shifts in floral composition occurred elsewhere in the world.

**Methods**

**Fossiliferous localities.** Samples were collected from the shallow-marine Río Turbio
Formation in southern Patagonia (Fig. 1). ~~The Río Turbio Formation preserves a thick~~
~~shallow marine and estuarine succession characterized by sandstones, limestones and~~
~~conglomerates interbedded with clay horizons accumulated in coastal marine, wave and tide-~~
~~dominated shallow water environments~~ (15, 16). Our **high-resolution record** encompasses the
MECO as well as pre- and post-MECO floras. For the first time, we quantified plant species
richness in the southernmost regions of South America during the globally warm mid Eocene
~~Epoch~~ using palynological data (pollen and spores). Other fossils preserved in these deposits
include terrestrial (leaves (e.g. 17), trunks (e.g. 18)) and marine (shells (e.g. 19), foraminifera
(e.g. 20)) and dinocysts (e.g. 21)) remains. The spore-pollen bearing sediments are
~~constrained as the~~ mid-late Eocene (42–33.5 Myr) based on foraminifera (22) and
dinoflagellate cyst (e.g. 21; this study (see Supplementary Note 2)) data. Here, we studied
pollen samples collected from a *ca* 400 m section spanning approximately 10 million years of
the Eocene in order to better understand the effects of climatic change on continental biotas.

**Palynology.** A total of 53 samples from the Río Turbio Formation were processed.
Palynomorph and dinocyst preparations were undertaken at the Museo Argentino de Ciencias
Naturales and followed a basic procedure of maceration, chemical digestion of silicates
(hydrofluoric acid), fluorosilicates (chlorhidric acid), and a very light oxidation to remove
excess amorphous matter (2 minutes in 70% nitric acid). Finally, residues were concentrated
and mounted onto slides. Residues were sieved with 25 µm and 10 µm meshes. Fiftyone
samples yielded abundant pollen, spores and dinoflagellate cysts. Slides were scanned under
a transmitted light microscope Leica DM 500. Spores, pollen grains, and dinocysts were
photographed by a Leica camera ICC50 HD. A mean of 354 spores and pollen grains and 285

dinocysts were counted per sample. Slides are housed at the Museo “Padre Jesús Molina”
under the catalogue numbers 21647–21699, prefixed MPM-PB. We removed some of the
samples (e.g. coal seam samples) for the subsequent biodiversity and abundance analyses,
particularly those that we interpreted as having been deposited in a more continental
paleoenvironment. Mudstones and fine sandstones preserving relatively high frequencies of
dinocysts (representing temporary marine incursions) contain significantly higher proportions
of pollen from wind-pollinated families (e.g. podocarps and southern beeches), which may
have originated a considerable distance inland from the coast. These samples therefore
represent a much larger source area (regional to sub-continental), compared to the local signal
contained in the coal measure samples, as previously tested (23).

**Quantitative analysis.** We conducted all analyses using the open-source software R (24). We
arranged pollen data in a 29 x 118 matrix with samples and taxa in which each cell contained
count data for all taxa of the selected samples (Supplementary Table 2).
We conducted a cluster analysis to explore sample associations (Q-mode) based on both
marine (~~dinocysts~~) and continental (~~pollen and spores~~) palynomorphs. We used the ‘chclust’
function of R/rioja (25) that performs a constrained cluster analysis of a distance matrix, with
clusters constrained by sample order. The distance matrix used was the Bray–Curtis metric
(26) and the agglomeration method was the CONISS (25).
For estimating biodiversity, we use three methods: 1) raw species richness (or ‘within flora
diversity’ *sensu* Lupia et al. 27), estimating the percentage of species belonging to each
sample relative to the diversity of the entire assemblage; 2) standing diversity (28) using the
range-through method which assumes that a taxon is present in a sample if it is found both
above and below it, and hence it tends to minimize facies effect; 3) rarefaction (29), ‘vegan’
R package, from relative abundance data of fossil spore–pollen assemblages, to estimate
species diversity relative to sample size. We use rarefaction because the count sizes for the

samples slightly differed. Differential count sizes will bias any estimate of within-sample
diversity as larger count sizes correspond to greater richness within a sample. Rarefaction
allowed us to down-sample those larger samples until they are the same size as the smallest
sample, making fair comparisons between incomplete samples. We contrasted our diversity
estimates from groups of samples defined by our dinocyst cluster analysis using boxplots (R
package ‘ggplot2’ (30)), and quantified differences using a Tukey’s test to identify which
groups were statistically different from each other.

**Data availability**

Slides are housed at the Museo “Padre Jesús Molina” under the catalogue numbers 21647-
21699, prefixed MPM-PB. The authors declare that the data that support the findings of this
study are available within this paper and its supplementary information files, and are
available from the corresponding author on reasonable request.

**References**

- 1. Bijl, P. K., Houben, A. J., Schouten, S., Bohaty, S. M., Sluijs, A., Reichert, G.,
Damsté J. S. & Henk Brinkhuis, H. Transient Middle Eocene atmospheric CO₂ and
temperature variations. *Science* **330**, 819–821 (2010).
- 2. Bohaty, S. M. & Zachos, J. C. Significant Southern Ocean warming event in the late
middle Eocene. *Geology* **31**, 1017–1020 (2003).
- 3. Bohaty, S. M., Zachos, J. C., Florindo, F. & Delaney, M. L. Coupled greenhouse
warming and deep-sea acidification in the middle Eocene. *Paleoceanography* **24**,
2207 (2009).
- 4. Jaramillo, C., Ochoa, D., Contreras, L., Pagani, M., Carvajal-Ortiz, H., Pratt, L. M.,
Krishnan, S., Cardona, A., Romero, M., Quiroz, L. & Rodriguez, G. Effects of rapid

- global warming at the Paleocene-Eocene boundary on neotropical vegetation. *Science*
**330**, 957–961 (2010).
- 5. Wilf, P., Johnson, K. R., Cuneo, N. R., Smith, M. E., Singer, B. S. & Gandolfo, M. A.
Eocene plant diversity at Laguna del Hunco and Río Pichileufú, Patagonia, Argentina.
*The American Naturalist* **165**, 634–650 (2005).
- 6. Exon, N.F., Kennett, J.P., Malone, M.J. in *Proceedings of the Ocean Drilling*
*Program, Initial Reports* (eds Exon, N.F., Kennett, J.P., Malone, M.J.) Volume 189.
[http://www.odp.tamu.edu/publications/189_IR/VOLUME/CHAPTERS/IR189_05.PD](http://www.odp.tamu.edu/publications/189_IR/VOLUME/CHAPTERS/IR189_05.PDF)
[F](http://www.odp.tamu.edu/publications/189_IR/VOLUME/CHAPTERS/IR189_05.PDF) (2001).
- 7. Cramwinckel, M. J., Woelders, L., Hurdeman, E. P., Peterse, F., Gallagher, S. J.,
Pross, J., Burgess, C. E., Reichert, G. J., Sluijs, A. & Bijl, P. K. Surface-circulation
change in the Southern Ocean across the Middle Eocene Climatic Optimum:
inferences from dinoflagellate cysts and biomarker paleothermometry. *Clim. Past*
*Discuss.* (2019). <https://doi.org/10.5194/cp-2019-35>.
- 8. Woodburne, M. O., Goin F. J., Bond M., Carlini A. A., Gelfo J. N., López G. M.,
Iglesias A. & Zimicz A. N. Paleogene land mammal faunas of South America; a
response to global climatic changes and indigenous floral diversity. *Journal of*
*mammalian Evolution* **21**, 1–73 (2014).
- 9. Jaramillo, C., Rueda, M. J. & Mora, G. Cenozoic plant diversity in the Neotropics.
*Science* **311**, 1893–1896 (2006).
- 10. Harrington, G. J. & Jaramillo, C. A. Paratropical floral extinction in the Late
Palaeocene–early Eocene. *Journal of the Geological Society* **164**, 323–332 (2007).
- 11. Bellosi, E. S. & Gonzalez, M. G. in *The Paleontology of Gran Barranca: evolution*
*and environmental change through the Middle Cenozoic of Patagonia* (eds Madden,

- R. H., Carlini, A. A., Vucetich, M. G. & Kay, R. F.) 293–305 (Cambridge University
Press, Cambridge, 2010).
- 12. Methner, K., Mulch, A., Fiebig, J., Wacker, U., Gerdes, A., Graham, S. A. &
Chamberlain, C. P. Rapid middle Eocene temperature change in western North
America. *Earth and Planetary Science Letters* **450**, 132–139 (2016).
- 13. Bosboom, R., Dupont-Nivet, G., Grothe, A., Brinkhuis, H., Villa, G., Mandic, O.,
Stoica, M., Huang, W., Yang, W., Guo, Z. & Krijgsman, W. Linking Tarim Basin sea
retreat (west China) and Asian aridification in the late Eocene. *Basin Research* **26**,
621–640 (2014).
- 14. Mancini, M. V. Vegetation and climate during the Holocene in Southwest Patagonia,
Argentina. *Review of Palaeobotany and Palynology* **122**, 101–115 (2002).
- 15. Furque, G. & Caballé, M. Estudio geológico y geomorfológico de la cuenca superior
del Río Turbio, provincia de Santa Cruz. *Consejo Federal de Inversiones, Serie*
*Investigaciones aplicadas, Colección Hidrología subterránea* **6**, 8–39 (1993).
- 16. Rodríguez Raising, E. M. Estratigrafía secuencial de los depósitos marinos y
continentales del Eoceno-Oligoceno temprano de la cuenca Austral, suroeste de la
provincia de Santa Cruz. (Universidad Nacional del Sur, Bahía Blanca, 2010).
- 17. Panti, C. Fossil leaves of subtropical lineages in the Eocene–? Oligocene of southern
Patagonia. *Historical Biology* **32**, 291–306 (2020).
- 18. Pujana, R. R. & Ruiz, D. P. Fossil woods from the Eocene–Oligocene (Río Turbio
Formation) of southwestern Patagonia (Santa Cruz province, Argentina). *IAWA*
*Journal* **1**, 1–31 (2019).
- 19. Griffin, M. Eocene bivalves from the Río Turbio formation, southwestern Patagonia
(Argentina). *Journal of Paleontology* **65**, 119–146 (1991).

- 20. Malumián, N. & Caramés, A. Upper Campanian-Paleogene from the Río Turbio coal
measures in southern Argentina: micropaleontology and the Paleocene/Eocene
boundary. *Journal of South American Earth Sciences* **10**,189–201 (1997).
- 21. González Estebenet, M. S., Guerstein, G. R., Rodríguez Raising, M., Ponce J.
&Alperín M. I. Dinoflagellate cyst zonation for the middle to upper Eocene in the
Austral Basin, Southwest Atlantic Ocean - Implications for regional and global
correlation. *Geological Magazine* **154**, 1022–1036 (2016).
- 22. Malumián, N. & Nández, C. in *Geología y Recursos Naturales de Santa Cruz* (ed
Haller M. J.) 481–94 (*Relatorio XV Congreso Geológico Argentino, El*
*Calafate*, 2011).
- 23. Jardine, P. E. & Harrington, G. J. The Red Hills Mine palynoflora: A diverse swamp
assemblage from the Late Paleocene of Mississippi, USA. *Palynology* **32**, 183–204
(2008).
- 24. Development Core Team. The R Project for Statistical Computing. Retrieved from
<http://www.R-project.org> (2020).
- 25. Juggins, S. Rioja: analysis of quaternary science data. R package v.0.7-3.[WWW
document] URL <http://cran.r-project.org/package=rioja>. (2012).
- 26. Bray, J. R. & Curtis, J. T. An ordination of the upland forest communities of southern
Wisconsin. *Ecological monographs* **27**, 325–349 (1957).
- 27. Lupia, R., Lidgard, S. & Crane, P. R. Comparing palynological abundance and
diversity: implications for biotic replacement during the Cretaceous angiosperm
radiation. *Paleobiology* **25**, 305–340 (1999).
- 28. Boltovskoy, D. The range-through method and first-last appearance data in
paleontological surveys. *Journal of Paleontology* **62**, 157–159 (1988).

- 29. Sanders, H. L. Marine benthic diversity: a comparative study. *Am. Nat.* **102**, 243–282
(1968).
- 30. Wickham, H. *ggplot2: Elegant Graphics for Data Analysis*. Springer-Verlag New
York. ISBN 978-3-319-24277-4 (2016).
- 31. Wrenn, J. H. & Hart, G. F. Paleogene dinoflagellate cyst biostratigraphy of Seymour
Island, Antarctica. *Geological Society of America, Mem* **169**, 321–447 (1988).
- 32. Mao, S. & Mohr, B. A. R. Middle Eocene dinocysts from Bruce Bank (Scotia Sea,
Antarctica) and their paleoenvironmental and paleogeographic implications. *Review*
*of Palaeobotany and Palynology* **86**, 235–263 (1995).
- 33. Amenábar, C. R., Montes, M., Nozal, F. & Santillana, S. Dinoflagellate cysts of the
La Meseta Formation (middle to late Eocene), Antarctic Peninsula: implications for
biostratigraphy, palaeoceanography and palaeoenvironment. *Geological Magazine*
**157**, 351–366 (2019).

**Acknowledgements**

We thank O. Cardenas and S. Mirabelli for assistance with processing; A. González for
assistance with drawing; V. Guler for assistance with stratigraphy; M.S. Candel for assistance
with dinocyst taxonomy. This work was supported by Consejo Nacional de Investigaciones
Científicas y Técnicas (PIP 2014–0259) and Agencia Nacional de Investigaciones Científicas
y Técnicas (PICT 2017–0671).

**Author contributions**

D.A.F, L.P. and V.D.B. contributed with the measurement and description of the stratigraphic
sections and collection of samples for palynological analyses. D.A.F. performed the
palynological counting (spore-pollen and dinocysts) and palynological assemblage

description, captured palynomorphs images and designed the figures illustrating them.
D.A.F., L.P. and V.D.B. wrote all draft and revised manuscript versions which includes
original and revised text supplied by co-authors. L.P. contributed with quantitative analysis
and illustrations. M.S.G.E. contributed with the dinocyst and stratigraphic analysis and
illustrations. D.A.F., M.C.T. and V.D.B. contributed with the analysis of spore-pollen
botanical affinity and phytogeography. All authors read and contributed to the manuscript.

**Competing interests**

The authors declare no competing interests.

**Correspondence** and requests for materials should be addressed to L.P.

**Figures**

**Fig. 1.** Location map showing distribution of Eocene sedimentary rocks of the Río Turbio
Formation, Santa Cruz Province, Patagonia, southern South America.

**Fig. 2.** Trends in frequency of ~~our marine components (i.e. dinocysts)~~. We identified three
major biogeographic groups in southern Patagonia: *Enneadocysta dictyostila*, E=Endemics
(e.g. *Vozzhennikovia apertura*, *Spinidinium macmurdoense*, *Deflandrea antarctica*,
*Arachnodinium antarcticum*, *Enneadocysta brevistila*, *Impletosphaeridium clavus*),
C=Cosmopolitan (e.g. *Turbiosphaera filosa*, *Thalassiphora pelagica*, *Spiniferites* spp.
*Operculodinium* spp., *Hystrichosphaeridium truswelliae*). The response of these
biogeographic groups from southern Patagonia to the MECO show a very close similarity
with that previously reported in Australia (6, 7), and Antarctica (31, 32, 33) (see
Supplementary Note 2 for further information). Our continental ~~components (spores and~~
~~pollen grains)~~ show a gradual increase in standing diversity during this warmth Phase B.

**Fig. 3.** Box-plots showing the standing diversity estimates for the three phases recognized by
our cluster analysis. Note that **Phase B** (MECO) is significantly different (Tukey's test, P
<0.0005) from **Phases A** (Pre-MECO) and C (Post-MECO), while these last two are not
different from each other. These differences are also supported by adjusted for sample size
(rarefaction) estimations and within-flora diversity (Table 1).

**Fig. 4.** Selected ~~spore-pollen~~ species from the Eocene of southern South America. 1–6.
Gondwanic elements (blue square); 7–12. Tropical elements (orange square). 1.
*Cyatheacidites annulatus*, sample 29 V18(1); 2. *Araucariacites australis*, sample 11 J44(2);
3. *Podocarpidites elegans*, sample 25 N41(4); 4. *Nothofagidites rocaensis*, sample 14 E40(1);
5. *Myrtaceidites verrucosus*, sample 11 H38(3); 6. *Propylipollis pseudomoides*, sample 16
C22(1); 7. *Ilexpollenites anguloclavatus*, sample 4 N57(4); 8. *Perisyncolporites pokornyi*,
sample 18 R16(3); 9. *Bombacacidites isoreticulatus*, sample 7 Y39(2); 10. *Cupaneidites*
*insulares*, sample 16 D13(4); 11. *Arecipites minutiscabratus*, sample 18 F47(3); 12.
*Monosulsites perspinosus*, sample 14 T37(4). Scale bar is 10 µm. Taxonomic names are
followed by the slide number and England Finder graticule coordinates.

**Table 1.** Diversity estimation of spore-pollen assemblage (i.e. Rarefaction, within-flora
diversity and standing diversity) for our **Phase A** (pre-MECO), **Phase B** (subgroups 1 to 3,
MECO), and **Phase C** (post-MECO).

Impact of mid Eocene greenhouse warming on America's southernmost floras

Fernández et al.

Table of Contents

**Supplementary Note 1: Stratigraphy and previous paleobotanical studies**

**Supplementary Note 2: Age model based on dinocyst assemblages**

**Supplementary Note 3: ~~Spore-pollen~~ assemblages**

**Supplementary References**

**Supplementary Figure 1.** Schematic correlation of the Río Turbio Formation showing
sections and sample locations.

**Supplementary Figure 2.** Selected dinocysts species from the Eocene of southern South
America.

**Supplementary Figure 3.** Quantitative distribution of the dinocyst assemblages from the Río
Turbio Formation.

**Supplementary Figure 4.** Cluster analysis of the continental palynomorphs along with the
frequency of the major plant groups.

**Supplementary Figure 5.** Frequency of selected southern gondwanan and tropical families.

**Supplementary Figure 6.** Dinocyst events and zones recorded in the Río Turbio Formation.

**Supplementary Table 1.** Taxonomic list of ~~species of dinocysts~~
Formation and ~~its~~ latitudinal distribution.

**Supplementary Table 2.** Species list, botanical affinity, distribution and ~~abundance data of~~
~~the spore-pollen assemblages~~.

Supplementary Information

Impact of mid Eocene greenhouse warming on America's southernmost floras

Supplementary Note 1: Stratigraphy and previous paleobotanical studies

The Río Turbio Formation (1, 2) comprises approximately 550-600 m of ~~marine, transitional~~

~~and terrestrial sediments, representing shallow-marine and estuarine successions.~~ Two

stratigraphically distinct (informal) members were recognized; a lower member (LM) ~~made up~~

~~of about~~ 290 m of conglomerates and coarse- to medium-grained sandstones, assigned to the

early to middle Eocene (3, 4) ~~and an~~ upper member (UM) of 300 m of fine to coarse sandstones

and conglomerates with interbedded clay horizons, representing coastal marine, wave- and

tide-dominated environments, assigned to middle ~~to~~ late Eocene (5). Most of the samples

collected for the present study comes from the UM (Supplementary Fig. 3). Both the LM and

UM contain coal seams of up to 2 m thick with abundant fossil plant remains. The Río Turbio

Formation conformably overlies the late Cretaceous Cerro Dorotea Formation (3, 5) and it is

unconformably overlain by the Miocene Río Guillermo Formation (6–8). Fossil remains either

of plants (leaves and woods) and spore-pollen and dinocyst assemblages preserved at the Río

Turbio Formation have been studied in detail since the last several decades (9–19). Although

the diversity estimates have never been quantified so far for the Río Turbio Formation on the

basis of the spore-pollen record, these studies constrained the age of the sedimentary unit, and

provided a general picture of what landscapes looked like during the Paleogene in southern

South America.

Supplementary Note 2: Age model based on dinocyst assemblages

The dinocyst stratigraphy of the Río Turbio Formation (this study) is largely based on the

magneto ~~and~~ chemo-stratigraphically calibrated middle–late Eocene South Pacific Dinocyst

Zones (SPDZ; 20) and the Dinocyst Associations (DA; 21), as well as a previous zonation
scheme for the Río Turbio Formation (RTF; 22–24). In general, the dinocyst assemblages from
the Río Turbio Formation are low diverse, mostly composed by the endemic Antarctic taxa
*Deflandrea* spp., *Vozzhennikovia apertura*, *Enneadocysta dictyostila*, and the cosmopolitan
*Turbiosphaera filosa*, *Operculodinium* spp. and *Spiniferites* spp., comprising more than 90%
of the total dinocysts (Fig. 2; Supplementary Fig. 3). The dinocyst assemblages from Phase A
(samples 1 to 4) contain frequent (up to 20 specimens) *Enneadocysta dictyostila*, while Phase
B (samples 5 to 22) present peak relative abundances of this species reaching 95% (up to 1026
specimens) associated with *Hystrichosphaeridium truswelliae* and *Arachnodinium antarcticum*
(Highest Occurrences, HOs; 36 Myr; 20). High abundances of *Enneadocysta dictyostila*
previously recorded in the Río Turbio Formation and the coetaneous upper Man Aike
Formation in the South of Lake Argentino, were linked to $^{87}\text{Sr} / ^{86}\text{Sr}$ ages between ~42 and 39
63 Myr (25). Furthermore, peaks of *E. dictyostila* (24) are associated with *Dracodinium*
*rhomboideum* (as *Rhombodinium* sp.; 24, 26), a stratigraphic MECO marker with its FO at
40.00 ± 0.10 Myr (20, 27). In Phase B we also recognize the upper part of the dinocyst Zone
RTF2 (~ 46 Myr – 39 Myr; 24), which correlate with the upper part of the dinocyst Zone
SPDZ12 (44 Myr – 40 Myr; 20) and the bottom of the Zone SPDZ13 (40.0 Myr–35.95 Myr;
20), spanning over the Middle Eocene Climatic Optimum (MECO) (Supplementary Fig. 6).
Thus, Phase B in the Río Turbio Formation comprises between ~40 and 39 Myr and the peaks
of *E. dictyostila* herein recognized would be related to the MECO. During the Middle-Eocene
in the southernmost gondwanan continents (i.e. paleo-latitudes south of 60° S), Antarctic-
endemic taxa are dominant in the dinocyst assemblages (e.g. 20, 28–38). However, prominent
changes in the dinocyst compositions denote biotic response to warming during the MECO at
~ 40 Myr (e.g. 20, 34, 39, 40). In the East Tasman Plateau (ODP Site 1172) an incursion of
low-latitude (cosmopolitan) dinocysts temporarily replaced the endemic taxa, and assemblages

largely consist of ~~the cosmopolitan species~~ *Enneadocysta multicornuta* (top of Zone SPDZ12;
20). Instead, in the South Tasman Rise in the central Tasman Gateway (ODP Site 1170), peaks
of *Enneadocysta dictyostila* were recorded during the MECO (32, 40). Moreover, peaks of
*Enneadocysta dictyostila* were also recorded in the middle Eocene (RTF2 Zone) of the Austral
Basin (e.g. 22–25, 41) and Antarctic Peninsula (38) associated with the global hyper thermal
episode (MECO) (24).

Finally, the Phase C (samples 23 to 29) contains peaks of *Vozzhennikovia apertura*
(lower part) and *Turbiosphaera filosa* (upper part) (Supplementary Fig. 3). The peak relative
abundances of *V. apertura* allows us to recognize the Late Bartonian to middle Priabonian Zone
RTF 3 (~ 39 Myr – 36 Myr; 24) that correlate with mid to upper part of the Zone SPDZ 13 (20)
(Supplementary Fig. 6). The dominance of *Turbiosphaera filosa* (~~lowest common occurrences:~~
~35.5 Myr; 21) together with the HOs of *Turbiosphaera filosa*, *Spinidinium macmurdoense*
and *Vozzhennikovia* spp. (HOs: ~33.5 Myr; 21, 42) in the upper part of the Phase C allow us to
identify the Zone RTF 4 of middle to late Priabonian age (~35.5–33.5 Myr; 24) which correlate
with the Dinocyst Association (DA) 2 of the South Pacific Ocean (~35.5–33.5 Myr; 21).
Towards the top of Phase C, typical endemic Antarctic assemblages were gradually replaced
by more cosmopolitan dinocyst taxa, reflecting changes in the oceanographic and
paleoenvironmental conditions related to the Tasman Gateway deepening (~35.5 Myr; 21, 35,
43) and the opening of the Drake Passage (37).

**Supplementary Note 3: Spore-pollen assemblages**

Our collected samples yielded moderate to large numbers of spores, pollen and dinocysts, in a
matrix of well-preserved to carbonized plant debris (including cuticles), finely disseminated
charcoal particles and amorphous algal and fungal remains. ~~The material recovered is relatively~~
~~well-preserved, making identification at morphospecies level possible for most of the spore-~~

~~pollen and dinocyst elements~~. The presence of tetrads (e.g. *Ericipites* sp. 1, *Bysmapollis*
*verrucatus*) and clusters of pollen grains (e.g. *Nothofagidites*) suggest **low-energy conditions**
at the time of the accumulation of the RTF.

Overall, the **terrestrial** palynological assemblages from RTF are dominated by
sporomorphs (~60 %), with abundant dinocysts (~40 %) and some acritarchs and acari claws.
The sporomorph assemblage consists of abundant angiosperm ~~pollen~~ (25–80 %) and ferns
spores (10–50 %), ~~with gymnosperm pollen as a minor component of the assemblage (10%)~~.
Angiosperm pollen are dominated by *Nothofagidites* spp. (mainly *Nothofagus* sg. *Nothofagus*)
and *Myrtacidites* spp. (Myrtaceae, Myrtoideae), with *Granodiporites nebulosus*, *Proteacidites*
spp. and *Propylipollis* spp. (Proteaceae) as minor elements. Ferns ~~spores~~ are mainly represented
by ~~tree ferns~~ (Cyatheaceae/Dicksoniaceae and Osmundaceae) ~~which consist~~ mainly of
*Cyathidites minor* and *Trilites* spp. and to a lesser extent, of *Cyatheacidites annulatus* and
*Baculatisporites* spp. Saccate pollen are ~~mainly~~ represented by *Podocarpidites* spp. and
*Phyllocladidites mawsonii* (Podocarpaceae); other gymnosperm is *Araucariacites australis*
(Araucariaceae, Araucaria). Furthermore, ~~occurrences of tropical species~~, *Ilexpollenites* spp.
(Aquifoliaceae, *Ilex*), *Cupaneidites* spp. (Sapindaceae, *Cupania*), *Bombacacidites*
*isoreticulatus* (Malvaceae Bombacoidea), and five species of palms (Arecaeae) are recorded
(Fig. 4). **At the beginning of the MECO** ferns increase from ~30% to ~60% while angiosperms
decrease dramatically from ~70% to ~30%. Also Podocarpaceae increase from ~5 % to ~20%
and tropical groups from ~2 % to ~6%. **At the core of the MECO** ferns drop to a minimum,
while angiosperms become dominant (80%). Apart from the dominant lineages (i.e.
*Nothofagus*, Podocarpaceae and Dicksoniaceae), other ~~gondwanan~~ families (e.g. Myrtaceae
and Proteaceae) became important elements (Supplementary Fig. 5). Tropical lineages also
became common and diverse (~85%) across the MECO. At the end of the MECO (sub-group
3, samples 16–14) ferns rise again to maximum values (ca. 60%) while angiosperms and

tropical groups decrease at the same time. Post-MECO ferns decrease towards the top while
Podocarpaceae remains abundant and a drop of tropical forms diversity (~40%) is recorded
(Supplementary Fig. 5).

**Supplementary References**

- 1. Riccardi, A. C. & Rolleri, E. O. Cordillera patagónica austral. *Simposio de Geología*
*Regional Argentina* **2**, 1173–1306 (1980).
- 2. Furque, G. & Caballé, M. Estudio geológico y geomorfológico de la cuenca superior
del Río Turbio, provincia de Santa Cruz. *Consejo Federal de Inversiones, Serie*
*Investigaciones aplicadas, Colección Hidrología subterránea* **6**, 8–39 (1993).
- 3. Malumián, N. in *Geología y recursos naturales de Santa Cruz* (ed Haller M. J.) 237–
245 (Relatorio XV Congreso Geológico Argentino, Buenos Aires, 2002).
- 4. Guerstein, G. R. & Daners, G. Distribución de Enneadocysta (Dinoflagellata) en el
Paleógeno del Atlántico Sudoccidental: implicancias paleoceanográficas. *Ameghiniana*
**47**, 461–478 (2010b).
- 5. Malumián, N., Panza, J. & Parisi, C. Yacimiento Rio Turbio: Instituto de Geología y
Recursos Minerales SEGEMAR (Argentina) Carta Geológica 5172-III, escala,
1:250.000 (2000).
- 6. Leanza, A. F. in *Geología Regional Argentina* (ed. Leanza, A. F.) 689–706 (Academia
Nacional de Ciencias, Córdoba, 1972).
- 7. Arguijo, M. H. & Romero, E. J. Análisis Bioestratigráfico de Formaciones portadoras
de Taofloras Terciarias. *Actas VIII Congreso Geológico Argentino* **6**, 691–717 (1981).
- 8. Ramos, V. A. in *Geología y recursos naturales de Santa Cruz* (ed Haller M. J.) 365–387
(Relatorio XV Congreso Geológico Argentino, Buenos Aires, 2002).

- 9. Hünicken, M. Flora Terciaria de los Estratos de Río Turbio, Santa Cruz (Niveles
plantíferos del arroyo Santa Flavio). *Rev. Fac. Cienc. Exact., Fis. y Nat. Univ. Córdoba,*
*S. Cs. Nat.* **27**, 139–227 (1967).
- 10. Panti, C. Myrtaceae fossil leaves from the Río Turbio Formation (Middle Eocene),
Santa Cruz Province, Argentina. *Historical Biology* **28**, 459–469 (2016).
- 11. Panti, C. Southern beech (Nothofagaceae) fossil leaves from the Río Turbio Formation
(Eocene–? Oligocene), Santa Cruz Province, Argentina. *Revista del Museo Argentino*
*de Ciencias Naturales nueva serie* **21**, 69–85 (2019).
- 12. Panti, C. Fossil leaves of subtropical lineages in the Eocene–? Oligocene of southern
Patagonia. *Historical Biology* **32**, 291–306 (2020).
- 13. Archangelsky, S. On the genus *Tomaxellia* (Coniferae) from the Lower Cretaceous of
Patagonia (Argentina) and its male and female cones. *Botanical Journal of the Linnean*
*Society* **61**, 153–165 (1968).
- 14. Archangelsky, S. Estudio del paleomicroplancton de la Formación Río Turbio
(Eoceno), Provincia de Santa Cruz. *Ameghiniana* **6**, 181–218 (1969).
- 15. Archangelsky, S. & Fasola, A. Algunos elementos del Paleomicroplancton del
Terciario inferior de Patagonia (Argentina y Chile). *Revista del Museo de la Plata* **6**,
1–18 (1971).
- 16. Archangelsky, S. Esporas de la Formación Río Turbio (Eoceno) Provincia de Santa
Cruz. *Revista del Museo de la Plata* **6**, 65–100 (1972).
- 17. Romero, E. J. Polen fósil de Gimnospermas y Fagáceas de la Formación Río Turbio
(Eoceno), Santa Cruz, Argentina (FECIC, Buenos Aires, 1977).
- 18. Archangelsky, S. & Romero, E. J. Polen de gimnospermas (coníferas) del Cretácico
superior y Paleoceno de Patagonia. *Ameghiniana* **11**, 217–236 (1974a).

- 19. Archangelsky, S. & Romero, E. J. Los registros más antiguos del polen de *Nothofagus*
(Fagaceas) de Patagonia (Argentina y Chile). *Boletín de la Sociedad de Botánica de*
*México* **33**, 13–30 (1974b).
- 20. Bijl, P. K., Sluijs A. & Brinkhuis H. A magneto-and chemostratigraphically calibrated
dinoflagellate cyst zonation of the early Palaeogene South Pacific Ocean. *Earth-Science*
*Reviews* **124**, 1–31 (2013).
- 21. Sluijs, A., Brinkhuis, H., Stickley, C. E., Warnaar, J., Williams, G. L. & Fuller, M. in
*Proceedings of the Ocean Drilling Program, Scientific Results* (eds Exon, N. F.,
Kennett J. P. & Malone, M. J.) 1–42 (ODP, 2003).
- 22. González Estebenet, M. S., Guerstein, G. R. & Rodríguez Raising, M. E. Middle
Eocene Dinoflagellate cysts from Santa Cruz Province, Argentina: biostratigraphy and
paleoenvironment. *Review of Paleobotany and Palynology* **211**, 55–65 (2014a).
- 23. González Estebenet, M. S., Guerstein, G. R. & Casadío, S. Estudio bioestratigráfico y
paleoambiental de la Formation Río Turbio (Eoceno medio a tardío) en el sudoeste de
Patagonia (Argentina) basado en quistes de dinoflagelados. *Revista Brasileira de*
*Paleontología* **18**, 429–42 (2015).
- 24. González Estebenet, M. S., Guerstein, G. R., Rodriguez Raising, M. E., Ponce, J. J. &
Alperín, M. I. Dinoflagellate cyst zonation for the middle to upper Eocene in the Austral
Basin, southwestern Atlantic Ocean: implications for regional and global correlation.
*Geological Magazine* **154**, 1022–1036 (2016).
- 25. Guerstein, G. R., González Estebenet, M. S., Alperin M. I., Casadío S. A. &
Archangelsky S. Correlation and paleoenvironments of middle Paleogene marine beds
based on dinoflagellate cysts in southwestern Patagonia, Argentina. *Journal of South*
*American Earth Sciences* **52**, 166–178 (2014).

- 26. González Estebenet, M. S. Quistes de dinoflagelados del Eoceno del sudoeste de Santa
Cruz. Análisis bioestratigráfico y paleoambiental. (Universidad Nacional del Sur, Bahía
Blanca, 2015).
- 27. Eldrett, J. S. & Harding, I. C. Palynological analyses of Eocene to Oligocene sediments
from DSDP site 338, outer Vøring plateau. *Marine Micropaleontology* **73**, 226–240
(2009).
- 28. Wrenn, J. H. & Beckman, S. W. Maceral, total organic carbon, and palynological
analyses of Ross Ice Shelf Project Site J9 cores. *Science* **216**, 187–189 (1982).
- 29. Wrenn, J. H. & Hart, G. F. Paleogene dinoflagellate cyst biostratigraphy of Seymour
Island, Antarctica. *Geol.Soc. Am. Mem.* **169**, 321–447 (1988).
- 30. Cocozza, C. D. & Clarke, C. M. Eocene microplankton from La Meseta Formation,
northern Seymour Island. *Antarctic Science* **4**, 355–362 (1992).
- 31. Mao, S. & Mohr, B. A. R. Middle Eocene dinocysts from Bruce Bank (Scotia Sea,
Antarctica) and their paleoenvironmental and paleogeographic implications. *Review of*
*Palaeobotany and Palynology* **86**, 235–263 (1995).
- 32. Exon, N.F., Kennett, J.P., Malone, M.J., et al. Proceedings of the Ocean Drilling
Program, Initial Reports Volume 189 (2001).
- 33. Huber, M., Brinkhuis, H., Stickley, C. E., Döös, K., Sluijs, A., Warnaar, J.,
Schellenberg, S. A. & Williams, G. L. Eocene circulation of the Southern Ocean: Was
Antarctica kept warm by subtropical waters?. *Paleoceanography* **19**, 4026 (2004).
- 34. Bijl, P. K., Pross, J., Warnaar, J., Stickley, C. E., Huber, M., Guerin, R., Houben, A.
219 J. P., Sluijs, A., Visscher, H. & Brinkhuis, H. Environmental forcings of Paleogene
Southern Ocean dinoflagellate biogeography. *Paleoceanography* **26**, 1202 (2011).
- 35. Houben, A. J., Bijl, P. K., Pross, J., Bohaty, S. M., Passchier, S., Stickley, C. E., Röhl
U., Sugisaki S., Tauxe L., Flierdt T., Olney M., Sangiorgi F., Sluijs A., Escutia C.,

- Brinkhuis H., Expedition 318 Scientists.. Reorganization of Southern Ocean plankton
ecosystem at the onset of Antarctic glaciation. *Science* **340**, 341–344 (2013).
- 36. Douglas, P. M., Affek, H. P., Ivany, L. C., Houben, A. J., Sijp, W. P., Sluijs, A.,
Schouten S. & Pagani, M. Pronounced zonal heterogeneity in Eocene southern high-
latitude sea surface temperatures. *Proceedings of the National Academy of Sciences*
**111**, 6582–6587 (2014).
- 37. González Estebenet, M. S., Guerstein, G. R. & Alperin, M. I. Dinoflagellate cyst
distribution during the Middle Eocene in the Drake Passage area: paleoceanographic
implications. *Ameghiniana* **51**, 500–9 (2014b).
- 38. Amenábar, C. R., Montes, M., Nozal, F. & Santillana, S. Dinoflagellate cysts of the La
Meseta Formation (middle to late Eocene), Antarctic Peninsula: implications for
biostratigraphy, palaeoceanography and palaeoenvironment. *Geological Magazine*
**157**, 351–366 (2019).
- 39. Bijl, P. K., Houben, A. J., Schouten, S., Bohaty, S. M., Sluijs, A., Reichart, G., Damsté
237 J. S. & Henk Brinkhuis, H. Transient Middle Eocene atmospheric CO₂ and temperature
variations. *Science* **330**, 819–821 (2010).
- 40. Cramwinckel, M. J., Huber, M., Kocken, I. J., Agnini, C., Bijl, P. K., Bohaty, S. M.,
Frieling, J., Goldner, A., Hilgen, F. J., Kip, E. L. & Peterse, F. Synchronous tropical
and polar temperature evolution in the Eocene. *Nature* **559**, 382–386 (2018).
- 41. Guerstein, G. R., Guler, M. V., Williams, G. L., Fensome, R. A. & Chiesa, J. O. Mid
Palaeogene dinoflagellate cysts from Tierra del Fuego, Argentina: biostratigraphy and
palaeoenvironments. *Journal of Micropalaeontology* **27**, 75–94 (2008).
- 42. Brinkhuis, H., Sengers, S., Sluijs, A., Warnaar, J. & Williams, G. L. in *Proceedings of*
*the Ocean Drilling Program. Scientific Results* (eds Exon, N. F., Kennett, J. P. &
Malone, M. J.) 1–48 (2003).

43. Stickley, C. E., Brinkhuis, H., Schellenberg, S. A., Sluijs, A., Röhl, U., Fuller, M.,
Grauert, M., Huber, M., Warnaar, J. & Williams, G. L. Timing and nature of the
deepening of the Tasmanian Gateway. *Paleoceanography* **19**, 4027 (2004).

44. Williams, G. L., Fensome, R. A. & MacRae, R. A. The Lentin and Williams index of
fossil dinoflagellates. *American Association of Stratigraphic Palynologists*
*Contributions Series* **48** (2017).

Dear referees,

Thank you for revising our work 'Impact of mid Eocene greenhouse warming on America's southernmost floras'. We modified our manuscript, analyses, plots, and plates based on your comments; we do think the quality of our contribution improved enormously with your suggestions. Thank you very much for your time and effort.

In the following, we present a point-by-point response (in blue) to the each of the issues you have raised:

Reviewer #1 (Remarks to the Author):

The paper by Fernández et al focuses on palynofloras from the MECO of southern Patagonia, and finds differences in diversity and composition compared to the pre- and post- MECO phases. Overall the paper is well-written, the dataset is a valuable one (as the authors note, very little is known about the MECO in the terrestrial realm), and the authors take on some interesting questions that would be relevant to people from a range of disciplines beyond palynology.

Response: We thank Reviewer #1 for her/his comments.

I do have some concerns however which I think need addressing before this paper can be published. I've started off with the main, overarching points, and then more specific, minor issues follow:

Main issues

1. Most of the samples are from the MECO interval, and there's very little context above and below the event. It's therefore very hard to know just how remarkable this event is: how high is the diversity relative to background conditions, given that you only have four samples before the onset of the MECO? And how atypical is it to have these tropical elements present in the flora? Supplementary Figure 5 suggests that there were at least some tropical taxa beforehand, with a similar abundance to some of the MECO samples (especially those in subgroup 2, in the core of the MECO). Without some knowledge of the diversity and composition above and below this section I'm not convinced that the authors can properly contextualise their data, and really assess the impact of the MECO warming.

Response: We have now included three extra samples from the pre-MECO interval in order to better contextualize this warming event. Many tropical and subtropical angiosperms are found across the entire sampled composite section, as this Reviewer #1 properly emphasizes. However, some tropical angiosperms (e.g. *Ceiba*, *Cardiospermum*, *Trimenia*, and palms (*A. subverrucatus*)) and ferns (e.g. *Anemiaceae*, *Cnemidaria*, *Schizaceae*) are mainly recorded during the MECO. In this new version, we now mention that not all tropical species are not

restricted to the MECO. Also, we now include a dedicated section with comments and comparisons of all spore and pollen species that we selected as tropicals (**Supplementary Note 4: Selected fossil forms with tropical and subtropical affinity**).

2. I'm not entirely convinced by your choice of diversity measures. Standing/range through diversity always suffers from the Signor-Lipps effect, with taxon first and last appearances never representing true first and last appearances but occurring relatively later or earlier respectively. This means that the taxon ranges get bunched up in the middle of the time series, even if they extend all the way to the ends, making diversity appear artificially higher in the middle and lower at the edges. It's an even bigger problem with this sort of dataset that covers a relatively short time period, and where the taxa probably have their first and last occurrences below and above the studied section (it's not clear from the methods if the authors did anything to correct for this, such as extending the ranges of those taxa that are known to occur earlier or later in time all the way to the ends of the section before tabulating diversity). In your case it will mean that MECO diversity will be biased towards higher values, as shown in Figure 2.

Response: We revised our analyses, and now we removed the Standing/Range-through diversity method from our work as it clearly inflated the diversity in the middle part of the time series, as Reviewer #1 highlights. Please, thank her/him for flagging this up.

I'm also not sure I follow the within-flora diversity aspect: not only does it seem quite hard to interpret in any intuitive way (perhaps it would make more sense in a spatial analysis when the interest is in linking communities to a regional species pool or something?), I'm also not clear why the values in Table 1 are all in the 30%*s* while in Figure 3 they are 0.6 – 0.9. I also note that when Figure 3 is considered, phase B has a much higher within-flora diversity than phases A and C, but when it is broken down in Table 1 into 3 subgroups the difference between the phases becomes much less pronounced – is this an issue with having more samples in phase B versus A and C, which then gets evened out when the dataset is split into the 5 phases in Table 1? I think my fundamental issue is that I really don't understand what this metric is telling me, and it certainly doesn't correspond to 'Raw species richness' (line 199), at least as I understand the term. Both this and the range through richness also make no corrections for differences in sample size.

Response: We have also removed from the analysis within-flora diversity, as it makes no correction for differences in sample size, as Reviewer #1 mentioned.

Rarefied richness makes a lot more sense, and as with within-flora diversity the difference between the MECO and the pre-and post- phases is much less pronounced in Table 1 than the range through richness plot suggests in Figure 2, which again suggests that the impact of the MECO is overstated in the paper. I also encourage the authors to look at coverage-based rarefaction (SQS in John Alroy terms; see Chao and Jost 2012) because coverage is a better measure of sampling completeness than count size. It would also be useful to look at evenness, because changes in evenness through a section will give the appearance of changes in richness when sampling is incomplete (relatively more even samples will appear to have

higher richness), and possibly richness estimators as well because these appear to be a bit less affected by this evenness issue.

Response: We applied coverage-based rarefaction in this new version of our manuscript, following Reviewer #1 recommendations. Also, we estimated evenness (the more even samples were indeed the more diverse ones).

My suggestion therefore is to alter how the diversity data are measured and presented in the paper. Rather than showing just range through richness at the edge of a figure which is dominated by dinoflagellate data (Figure 2), I would have a dedicated figure showing within-sample rarefied richness against height in the section (with confidence intervals to show the error in the richness estimates), plus a suitable evenness metric and within-sample extrapolated richness (e.g. the Chao1 estimator) as well. The sporomorph cluster dendrogram that is currently relegated to the supplementary figures could also occur here. It might also be worth thinking about looking at ordinations, such as NMDS or correspondence analysis. If the MECO interval really is different in its composition, then this would be picked up more clearly than in the constrained cluster analysis (the PETM analysis in Wing et al 2005 gives a nice example of this kind of analysis).

Response: The diversity was re-estimated using within-sample rarefied richness (Table 1). We now have a dedicated figure showing the variations of diversity against our composite section (with confidence intervals) plus evenness along the sporomorph cluster dendrogram, as Reviewer #1 suggested (Fig 2). The Chao1 and NMDS analysis were carried out and plotted (Figs 3 and 8, respectively).

If you want to show within-phase richness to compare pre-MECO, MECO and post-MECO diversity, sample based rarefaction curves sensu Gotelli and Colwell (2001) would be a better way to show this than the box plot in Figure 3, and can be carried out either with sample size or coverage-based rarefaction (see Chao and Jost 2012). Species richness estimators can also be calculated and plotted in this way, e.g. the Chao2 estimator for comparing richness among groups of samples.

Response: The box plot analysis was now excluded from our work, and it was replaced by sample-based rarefaction curves to compare pre-MECO, MECO and post-MECO diversity (Fig 4).

There are lots of examples of these sorts of analyses, applied to sporomorph data, in the literature, see for example Harrington and Jaramillo (2007), Harrington et al (2004), Jaramillo et al (2010), Jardine et al (2018), Wing et al (2005), plus various other papers by the same authors.

I've written a lot here, but since the main thrust of your paper rests on the atypical nature of the MECO in terms of plant diversity and composition, you need to make sure you use appropriate analytical approaches so that these claims can be supported.

Response: We appreciate Reviewer #1 recommendations.

3. As it stands the paper is well under the limit for word count and number of display items that the journal allows, and I think it could really benefit from a bit more detail and careful

discussion, rather than going straight for a headline story that isn't fully supported (at least with the way the data are presented now). As noted above, more data figures would probably help here. In addition to the analyses I've mentioned, bringing in the sporomorph relative abundance plots from the supplementary information into the main paper would probably be useful, and if anything needs cutting to allow for this, I think the dinoflagellate data in Figure 2 and the sporomorph images in Figure 4 could go into the supplementary information instead.

Response: Now, we discussed the MECO context and the floristic response in further detail, and compared it with that recorded in lower latitudes. We have now brought the sporomorph relative abundance plot into the main paper and increased the number of figures. As we now have two sporomorph plates, we place one in the main manuscript and the other in the supplementary information.

4. I appreciate the inclusion of the sporomorph count data as a supplementary file, but it would be useful to have similar data for the dinoflagellates, rather than these just being listed by name in Supplementary Table 1. The R code you used to analyse your data would also be valuable, so that everything is fully reproducible.

Response: We have now included the sporomorph and dinoflagellate count data as supplementary files. The R codes are also included in the Supplementary file (Supplementary Note 5).

Minor points

Lines 26-28: I understand the value of paleo-records for forecasting Earth's future, but I'm not convinced this does provide a good analogue for near-future conditions: so much of your reconstruction depends on taxa shared with a vegetated Antarctic (or the Gondwanan continents more generally), and out of the tropics migration routes that probably wouldn't be available in today's fragmented habitats. I suspect the lessons from a 40 million year old record provide limited data for forming solid predictions about Earth's immediate future, and I suggest toning down this sentence a bit (impressive though it sounds).

Response: We have now expressed this sentence differently, as Reviewer #1 suggested.

Line 43: The floras in reference 5 (Wilf et al 2005) are early and middle Eocene but not PETM.

Response: Modified.

Lines 138-141: does this imply relatively wetter conditions at the onset and end of the MECO, as defined by subgroups A and C when ferns increase in abundance?

Response: We assume that relatively wetter conditions prevailed at the onset and end of the MECO. We have discussed this further to better contextualize our interpretation.

Line 146: I'm really not convinced by this comparison with extant floras. Holocene pollen samples are based around a different taxonomic concept to deeper time material (extant

genera and families versus morphologically-circumscribed form taxonomy) and so I don't think the two things are really comparable. I realise that you are following Jaramillo et al here, but I was sceptical of this in their paper too. I think this statement, and the one that follows on comparing Patagonian richness through time, really needs unpacking and justifying, with the rarefied richness estimates given in each case (graphically if you like) and the rationale for this comparison given.

Response: We have now removed the Holocene comparison from our work.

Line 149: And what are these major implications? If you want to say this then I think it needs at least a few sentences specifying what these are (this comes back to my earlier comment about the final sentence in the abstract).

Response: We have now modified the sentence.

Lines 182-190: this isn't vital here, but for future work it would be interesting to look at the coal samples in a bit more detail, to give more information on the local (swamp?) taxa in comparison to the regional vegetation. Getting whatever information you can from the cuticles (mentioned in supplementary note 3) would also be worthwhile.

Response: We appreciate Reviewer #1 comments. We have in mind comparing the floristic composition and diversity of local (coal samples with mainly continental palynomorphs and cuticles) with regional or extra-regional (mudstones with marine and continental palynomorphs) in the near future.

Supplementary figures 4 and 5 - are the taxon abundances as specimen counts? These would make more sense as percentages.

Response: We have now modified those figures accordingly.

References

Chao, A. & Jost, L. (2012) Coverage-based rarefaction and extrapolation: standardizing samples by completeness rather than size. *Ecology*, 93, 2533-2547.

Gotelli, N.J. & Colwell, R.K. (2001) Quantifying biodiversity: procedures and pitfalls in the measurement and comparison of species richness. *Ecology Letters*, 4, 379-391.

Harrington, G.J. & Jaramillo, C.A. (2007) Paratropical floral extinction in the Late Palaeocene-Early Eocene. *Journal of the Geological Society, London*, 164, 323-332.

Harrington, G.J., Kemp, S.J. & Koch, P.L. (2004) Palaeocene-Eocene paratropical floral change in North America: responses to climate change and plant immigration. *Journal of the Geological Society, London*, 161, 173-184.

Jaramillo, C.A., Ochoa, D., Contreras, L., Pagani, M., Carvajal -Ortiz, H., Pratt, L.M., . . . Vervoort, J. (2010) Effects of rapid global warming at the Paleocene-Eocene boundary on Neotropical vegetation. *Science*, 330, 957-961.

Jardine, P.E., Harrington, G.J., Sessa, J.A. & Dašková, J. (2018) Drivers and constraints on floral latitudinal diversification gradients. *Journal of Biogeography*, 45, 1408-1419.

Wing, S.L., Harrington, G.J., Smith, F.A., Bloch, J.I., Boyer, D.M. & Freeman, K.H. (2005) Transient floral change and rapid global warming at the Paleocene-Eocene boundary. *Science*, 310, 993-996.

Thanks anonymous referee for your revision.

Reviewer #2 (Remarks to the Author):

The Eocene of Patagonia is one of the most fascinating research topics, when studying the effects of climate change on vegetation. At the time South America was still connected to Antarctica, with temperatures fluctuating but also reaching the highest values recorded in the Cenozoic. There is a broad body of literature on the diverse Eocene Patagonian flora, in which the ‘mixed’ nature, Gondwanan/Antarctic and tropical composition, is highlighted (e.g. Wilf et al., 2003, 2005, Barreda & Palazzesi, 2007; Vento & Pramparo, 2018). Particularly Vento & Pramparo (and references therein) give a detailed account of the mixed character of the Patagonian Eocene flora as recorded in the Rio Turbio Fm and they also point at the importance of the MECO. What is less clear is how the transition of this flora into cooler climate is expressed.

Response: We thank Reviewer #2 for her comments. We have now briefly discussed the transition from the mid Eocene hyperthermal event to the subsequent cooler/dryer conditions that witnessed the demise of tropical elements from Patagonia.

In their paper Fernandez et al. document the Patagonian Rio Turbio Formation and provide a detailed log and new high resolution dating based on a dinoflagellate analysis. They also provide a quantitative sporomorph analysis that assesses changes in pollen composition and diversity, and a statistical analysis that follows the biotic response before, at, and after the MECO. The paper is very interesting, well written and includes clear synthetic figures. The real selling point of the paper is the quantification of the biotic changes across the important MECO climate transition in this low latitude location.

Response: We thank Reviewer #2 for her comments.

Having said all this, there are some important issues that need to be addressed before the paper can be considered for publication.

1- The identification of the ‘tropical’ taxa. Tropical is used here as one of the hooks of the paper and uses the presence of Bombacaceae, Arecaceae, Malpigiaceae, among others, as key

argument. However, the light microscopy photos of these taxa are not very convincing. The photo plate does not have a good resolution (maybe I have the low-resolution version?) and none of the ‘tropical’ taxa seem really tropical; the identifications are in some (critical) cases doubtful. The two Arecaceae (Fig. 4: photos 11 and 12), both could easily be fern spores, in particular the taxon identified as *Monosulcites perspinosus*. Nevertheless, *Arecipites minutiscabratus* also does look spore-like. Better images and identifications are necessary due to the important role that these taxa play in the paper!

Response: We have now re-assessed all illustrated fossil pollen and spore specimens, as some of them were misidentified as Reviewer #2 properly mentioned. We have included a new section in the Supplement (Supplementary Note 4) commenting on the morphology and distribution of the key tropical taxa in order to better support our discoveries. We also added a new plate with more photomicrographs in this new version of our manuscript.

Similarly, the taxon identified as *Bombacidites isoreticulatus* (photo 9), easily could represent a *Ceiba* (*speciosa* type?), a Bombacaceae that tolerates temperate climate and is present today in Argentina. In addition, the taxon labeled as *Perisyncolporites pokorny* (photo 8) does not have the characteristics of this fossil. I am aware that this species has been previously reported in the Rio Turbio Fm by the first author (in a different paper), so a more convincing photo should be included. In addition, the biogeography of *Cupania* and *Ilex* is not restricted to the tropics. It should be noted that it does not require a tropical climate to maintain some of the above taxa in Argentine. Did they adapt to a cooler climate after the Eocene? Maybe, but this should be discussed. Note for instance the paper on fossil macroremains of palm material in the Eocene by Romero in Ameghiana (1968) on *Palmoxylon patagonicum*, who says the following: “it is concluded that *P. patagonicum* has intermediate characters with the subfamilies Sabaloideae, Coccoideae and Bactrioideae, which are now represented in Chile and Argentina. On this ground it is suggested that *P. patagonicum* might belong to the ancestral stock from which these subfamilies evolved”.

Response: We thank Reviewer #2 comments. We have now revised all the specimens. Some of the tropical (or subtropical) taxa that we mentioned became locally extinct (probably pushed northward) from southern Argentina (Patagonia), but they are indeed represented in the neotropics of either Argentina or Brazil, among other countries. We have now included the distribution of our discoveries in our new dedicated Supplementary Note 4.

In the abstract (line 23) therefore, do not use “very close to that occurring today in the Neotropics”. Although at the time there is a connection/exchange with the tropics, that statement does not seem appropriate for the taxa that are listed.

Response: We have now modified the abstract accordingly. Although, it is worth noting that Argentina (due to its vast size and range of altitudes) possesses a wide variety of climatic regions, ranging from hot subtropical in the north to cold subantarctic in the far south.

Lines 110-112: the fern taxa that are listed are mountain ferns in the lower latitudes, some considered immigrants of southern origin in today’s northern South American mountains. They are not genera typical of the tropical lowlands.

Response: We have now modified the text accordingly; within Supplementary Note 4 (lines 141, 159, 171, 186, 199). We now included the affinity of the identified ferns.

2 - I miss a reflection on what already is known from this formation and how this new study contributes to existing knowledge. For instance, Guerstein et al. is referred to for the taxonomic composition. When comparing the dinoflagellate diagram in the Guerstein paper (which includes the Rio Turbio) with the current paper, it is virtually identical. How does present study add to this?

Response: We are presenting the response of the floras to the MECO event in southernmost latitudes of South America. The dinocyst analysis from our work is based on a new sampling strategy, very similar to that of Guerstein et al. (2014), and Gonzalez Estebenet et al (2016). In our work, we re-assessed, and revised the dinocyst assemblage based on these and other publications (e.g. Cranwickel, 2020) to better constrain the extent of the MECO. We have now modified the Supplementary Note 2 to better clarify this aspect.

Methods: lines 185-190 the sampling rationale is clear. The authors highlight that mudstones were selected to avoid local bias. However, is it not the case that mudstones will have bias toward wind-pollinated taxa? They may represent a larger area, but not necessarily the diversity of that area.

Response: We have now modified this part of the Methods as it was probably not clear enough (lines 200-206).

Figures: supplement the log in figure 1 should have the m-scale next to the first section so that we can see the thickness in meters right away. The tiny scale in the legend, and thickness of selected strata is not very helpful.

Response: We have now included the m-scale next to the first section.

Thanks Carina for your revision.

Carina Hoorn, 22nd of July 2020

Reviewer #3 (Remarks to the Author):

After maximum temperatures in the early Eocene, ensuing mid-late Eocene long-term gradual cooling ultimately lead to major expansion of Antarctic ice-sheets at the Eocene-Oligocene transition. This transitional period of cooling is characterized by a short-lived warming event – the Middle Eocene Climatic Optimum (MECO) that has been discovered in ocean drilling cores from the Atlantic, Pacific and Southern Oceans and in land-based marine sections in Italy and UK. The MECO is marked by a rapid negative shift in both oxygen and carbon isotope ratios and thought to reflect an increase in sea surface and bottom water temperatures by up 5-6°C. However, well-dated high-resolution records encompassing the MECO are still

lacking in terrestrial realm and it is still largely unknown how the MECO-event affected the vegetation dynamics (i.e. diversity increase, plant migration patterns etc.).

From this point of view, the present manuscript of Fernandez et al. entitled “Impact of mid Eocene greenhouse warming on America’s southernmost floras” is an important and very interesting paper dealing with the terrestrial paleoecological repercussions of the MECO in southernmost South America. Based on new quantitative (including cluster analysis) palynological results from a number of sections from Patagonia (famous Rio Turbio Formation), this publication represents a first evidence for extraordinary plant diversity during the MECO as a response to this greenhouse event in high austral latitudes. According to author’s data, three intervals of different flora’s diversity have been recognized at the beginning, within and at the end of MECO, suggesting a massive turnover from extraordinary diverse mid-Eocene rain-forest biomes through the modern steppe-dominated landscapes.

Some drawback of the article could be called the lack of isotopic analyses confirming the MECO event directly in the Rio Turbio Formation. Nevertheless, it is known that in Southern Hemisphere the dinoflagellate cyst assemblages, calibrated by isotopic data to the MECO, are very specific and are characterized by the unique in the Eocene record dominance of endemic species *Enneadocysta dictyostila*. Consequently, taking into account a well-defined calibration of sporomorph assemblages from the marine sediments of the Rio Turbio Fm to the lowermost common occurrences of *Enneadocysta dictyostila*, this can serve as a basis for the assertion of the MECO event.

Response: We appreciate Reviewer #3 comments. Please, thank her on our behalf.

I think that the paper of Fernandez et al. merits to be published in “Communications Biology”. The presented results are novel and will be definitely of great interest to a large community of paleontologists, geologists and paleoclimatologists working on the Cenozoic. Statistical analysis is appropriate; conclusions are clear and convincing.

I have made a number of suggestions directly on the manuscript and supplementary information (pdf-file attached).

Here I just would like to make two comments of the Figures:

(1) Supplementary Figures 1 and 3:

The general stratigraphic scheme (Stages) and Dinocyst zones should be moved on the left of figure, while the Phases (I suggest to replace them in the text by Intervals) can be left on right.

Response: We moved to the left the figure, and used the term ‘Interval’ throughout the manuscript in this new version.

(2) Supplementary Figure 6: Please precise which column demonstrates the present study.

Response: We have now specified these data in both columns.

I hope that my review will help.

Thanks Alina for your revision.

Sincerely,

Alina I. Iakovleva

Geological Institute, Russian Academy of Sciences, Moscow, Russia.

REVIEWERS' COMMENTS:

Reviewer #1 (Remarks to the Author):

I thank the authors for their thorough revision of the manuscript, and for addressing both my comments and those of the other reviewers so diligently. I think this is an excellent piece of research, and I look forward to seeing it published in the very near future. I only picked up on three exceptionally minor points in the text which I suggest revising:

Line 108: suggest 'widely distributed throughout the MECO'

Line 137: suggest replacing 'assuming' with 'suggesting'

Line 171: I'm not sure what you mean by 'diversity event'

Reviewer #2 (Remarks to the Author):

Dear Authors,

I am very pleased to see the revised version of the Rio Turbio manuscript. Here some final comments:

1) Please add the ages and the depth/meter scale next to all of the graphs that show the palynology through time (fig. 2, 3, 5, 6). Also please add the dinocyst events next to the (new) columns with age and depths. I appreciate that the ages are shown in the supplement [Supplementary Figure 5], but it is really important to see it next to the data that are central to the paper. The figures cannot be published without this information (preferably also add the formation name).

2) Species nr 12 [Supplementary Figure 6] looks like a palm, but species nr 11 named *Arecipites minutiscabratus*, in Fig 8 still looks very much like a spore to me (as I mentioned in the first round of reviews). That is quite a crucial difference in this paper. I would suggest to replace '11' with the nr '12' from the supplement if you want to show a certain palm pollen grain.

3) Note that in the supplement some of the genus & species names are glued together, please add spaces where needed. Probably due to file conversion?

Dear Editor,

Below, is a point-by-point response to all reviewers' comments.
Please, thank them on our behalf.

Kind Regards,

Luis Palazzesi

Reviewer #1 (Remarks to the Author):

I thank the authors for their thorough revision of the manuscript, and for addressing both my comments and those of the other reviewers so diligently. I think this is an excellent piece of research, and I look forward to seeing it published in the very near future. I only picked up on three exceptionally minor points in the text which I suggest revising:

Response: We thank Reviewer #1 comments. We really appreciate her/his effort and time in improving our work.

Line 108: suggest 'widely distributed throughout the MECO'

Response: Modified.

Line 137: suggest replacing 'assuming' with 'suggesting'

Response: Modified.

Line 171: I'm not sure what you mean by 'diversity event'

Modified: We have now modified this sentence.

Reviewer #2 (Remarks to the Author):

Dear Authors,

I am very pleased to see the revised version of the Rio Turbio manuscript. Here some final comments:

1) Please add the ages and the depth/meter scale next to all of the graphs that show the palynology through time (fig. 2, 3, 5, 6). Also please add the dinocyst events next to the (new) columns with age and depths. I appreciate that the ages are shown in the supplement [Supplementary Figure 5], but it is really important to see it next to the data that are central to the paper. The figures cannot be published without this information (preferably also add the formation name).

Response: We have now included the time (Epoch and Stages) along the requested figures. Also, the dinocyst event next to the columns (sedimentary sections) have also been provided

in Figures 2 and 3. We appreciate Reviewer#2 for flagging this up.

2) Species nr 12 [Supplementary Figure 6] looks like a palm, but species nr 11 named *Arecipites minutiscabratus*, in Fig 8 still looks very much like a spore to me (as I mentioned in the first round of reviews). That is quite a crucial difference in this paper. I would suggest to replace '11' with the nr '12' from the supplement if you want to show a certain palm pollen grain.

Response: We removed *A. minutiscabratus* as it looks like a spore, as Reviewer #2 mentioned. And replaced it following her suggestion.

3) Note that in the supplement some of the genus & species names are glued together, please add spaces where needed. Probably due to file conversion?

Response: We have now revised the entire Supplementary Section and fixed the genus & species names as many of them were glued together. Please, thanks her on our behalf.